# Investigating the impact of poverty on mental illness in the UK Biobank using Mendelian randomization

**Mattia Marchi** [1,2,3], **Anne Alkema**[3], **Charley Xia** [4,5], **Chris H. L. Thio** [6,10],
**Li-Yu Chen**[3], **Winni Schalkwijk** [3], **Gian M. Galeazzi**[1,11] ✉, **Silvia Ferrari**[1,2],
**Luca Pingani**[1,2], **Hyeokmoon Kweon** [7], **Sara Evans-Lacko** [8], **W. David Hill** [4,5] &
**Marco P. Boks** [3,9,12] ✉

It is unclear whether poverty and mental illness are causally related. Using UK Biobank and Psychiatric Genomic Consortium data, we examined evidence of causal links between poverty and nine mental illnesses (attention deficit and hyperactivity disorder (ADHD), anorexia nervosa, anxiety disorder, autism spectrum disorder, bipolar disorder, major depressive disorder, obsessive-compulsive disorder, post-traumatic stress disorder and schizophrenia). We applied genomic structural equation modelling to derive a poverty common factor from household income, occupational income and social deprivation. Then, using Mendelian randomization, we found evidence that schizophrenia and ADHD causally contribute to poverty, while poverty contributes to major depressive disorder and schizophrenia but decreases the risk of anorexia nervosa. Poverty may also contribute to ADHD, albeit with uncertainty due to unbalanced pleiotropy. The effects of poverty were reduced by approximately 30% when we adjusted for cognitive ability. Further investigations of the bidirectional relationships between poverty and mental illness are warranted, as they may inform efforts to improve mental health for all.

The association between mental illness and social class was first demonstrated in a 1958 study by Hollingshead and Redlich, who found that individuals from lower socio-economic backgrounds had a higher incidence of severe and persistent mental illness and received less adequate treatment[1]. More than 50 years later, the same social conditions persist and affect mental health worldwide. Epidemiological studies throughout the world have demonstrated an association between mental health and socio-economic status (SES)[2–4], with mental illness being more common among people from lower social classes[5,6]. Also, studies on income fluctuations have found consistent changes in mental health[7,8].

Although the association between poverty and mental illness is strong across these studies, there is limited evidence that supports a causal relationship. Several factors, such as reverse causation and residual confounding, make it difficult to determine whether poverty causally contributes to mental illness or whether it is the other way around and mental illness leads to poverty. However, understanding the causality of the relationship between poverty and mental illness may be crucial for public health policies, as they may target essential aspects of poverty and improve public mental health[9].

To date, the direction of the association between poverty and mental illness remains uncertain, and two explanatory hypotheses compete: social causation and social selection[10]. According to the social causation theory, the socio-economic adversity faced by lower socio-economic groups precipitates mental illness in vulnerable

**Fig. 1 | Research design.** $\beta_1$ is the association between the genetic variants and the exposure of interest (MR relevance assumption), $i$ is the potential violation of the MR independence assumption, $p$ is the potential pleiotropic effect of the genetic variants on the outcome (that is, violation of the MR exclusion restriction assumption) and $\beta_2$ is the causal association of interest. Created with BioRender.com.

individuals, possibly mediated by factors such as housing insecurity, substance use and stress. Conversely, the social selection theory suggests that the overrepresentation of lower SES among people with mental illness is mainly attributable to downward social mobility as a consequence of the impairment associated with poor mental health[11].

Conducting randomized controlled trials to determine the causality of the role of poverty is neither feasible nor ethical. An alternative method to investigate is Mendelian randomization (MR), which uses genetic data from genome-wide association studies (GWAS) to examine whether a risk factor fits in a causal model for an outcome[12]. The MR method takes advantage of the fact that genetic variants are fixed at conception and are less susceptible to the confounding effects that make results from observational studies difficult to interpret[13].

Studies on poverty come with the challenge of defining poverty. Most of them use a single measure, usually based on income recorded at the household level, individual income or operationalized employment status of an individual, by occupational income[7,11]. However, no single indicator can capture the multiple dimensions of poverty, such as lack of material goods, limited access to education and health-care services, inadequate living standards, disempowerment, poor quality of work, the threat of violence, and living in areas that are environmentally hazardous, among others[14–17]. For that reason, the UK health authorities commonly use a composite measure known as the Townsend Deprivation Index to assess material deprivation within a population[18].

In this study, we applied MR to examine the causal effects of poverty on nine mental illnesses. First, to maximize the power required to examine the common effects of poverty across its multidimensional aspects, we derived a poverty factor using household income (HI), occupational income (OI) and social deprivation (SD). These three measures capture poverty at the levels of the individual, the household and the area in which one lives and so facilitate a more informed understanding of which aspects of poverty present the greatest risk to individual mental health. Second, we investigated the causal relationship of the common factor of poverty and each of the three indicators of poverty with nine mental illnesses: attention deficit and hyperactivity disorder (ADHD), anorexia nervosa (AN), anxiety disorder (ANX), autism spectrum disorder (ASD), bipolar disorder (BD), major depressive disorder (MDD), obsessive-compulsive disorder (OCD), post-traumatic stress disorder (PTSD) and schizophrenia (SZ). To account for potential confounding effects, we also extended the MR analyses to include cognitive ability (CA).

## Results

The schematic overview of the study is represented in Fig. 1.

### Latent-poverty-factor estimation

To analyse the joint genetic architecture of poverty, we ran a multivariable GWAS for which a common factor defined by genetic indicators is regressed on a single nucleotide polymorphism (SNP). The multivariable GWAS of the latent poverty factor (PF) was modelled from three indicators: HI, SD and OI (see Methods and Supplementary Information for details and Supplementary Table 1 showing the factor loading of each indicator). The heritability of the common factor GWAS was estimated with $h^2$ and was around 8.4%. The mean $\chi^2$, multivariable linkage disequilibrium score regression (LDSC) intercept and $h^2$ for the common factor GWAS and the GWAS of HI, SD and OI are presented in Supplementary Table 2.

The GWAS of the latent poverty factor identified 90 significantly associated independent loci at the GWAS threshold ($P \le 5 \times 10^{-8}$), as displayed in Fig. 2.

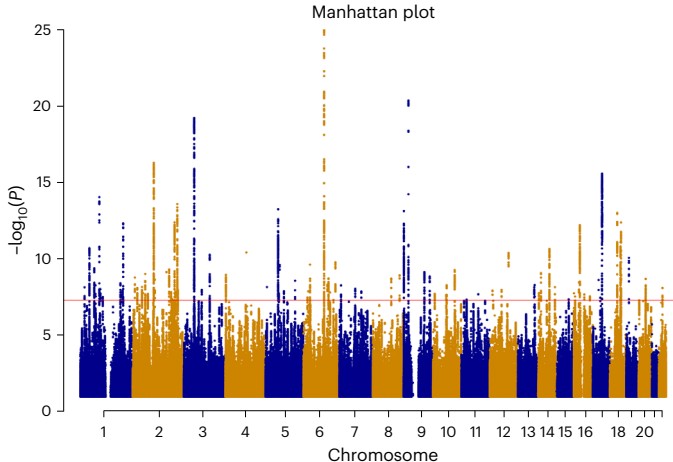

**Fig. 2 | Manhattan plot of the latent poverty factor.** The *x* axis indicates the chromosomal position, and the *y* axis indicates the significance of the association ($-\log_{10}(P)$). The red line represents the genome-wide significance level ($5 \times 10^{-8}$). SNP effects are estimated with a linear regression model. The *P* values are two-sided and not adjusted for multiple testing.

The genetic correlation between PF and CA was strong ($r_g = 0.74$, s.e. = 0.029; Supplementary Table 3). When we ran bidirectional MR of CA against PF, we found stronger evidence supporting the causal effect of CA on PF (inverse variance weighted (IVW) $\beta_{\text{CA}\rightarrow\text{PF}} = -0.390$ (95% confidence interval (CI): −0.408, −0.372)) rather than vice versa (IVW $\beta_{\text{PF}\rightarrow\text{CA}} = -0.274$ (95% CI: −0.288, −0.261)), as suggested by the results of Steger's test ($P < 0.001$ and $P = 0.423$, respectively). The full results are presented in Supplementary Table 4 and Supplementary Figs. 1–4.

### Bidirectional univariable MR of common factor poverty against mental illness

For the univariable analyses, the mean *F* statistic ranged from 31.5 to 45.6, indicating that the estimates were probably not subject to weak instrument bias. Forward IVW analysis of PF against the considered mental illnesses showed significant causal effects of PF on ADHD ($\beta = 0.330$ (95% CI: 0.287, 0.373)), ANX ($\beta = 0.229$ (95% CI: 0.101, 0.357)), MDD ($\beta = 0.115$ (95% CI: 0.074, 0.156)), PTSD ($\beta = 0.140$ (95% CI: 0.073, 0.207)) and SZ ($\beta = 0.110$ (95% CI: 0.074, 0.145)), and with the opposite direction on AN ($\beta = -0.192$ (95% CI: −0.254, −0.129)) and OCD ($\beta = -0.202$ (95% CI: −0.355, −0.049)). The estimate of the causal effect using the weighted median (WM) method was consistent in magnitude for ADHD, AN, MDD, PTSD and SZ, but not for ANX and OCD. We did not identify any significant causal effect of PF on BD or ASD.

Backward IVW analysis of mental illness against PF showed significant causal effects of ADHD ($\beta = 0.402$ (95% CI: 0.348, 0.455)), BD ($\beta = -0.091$ (95% CI: −0.133, −0.049)) and SZ ($\beta = 0.082$ (95% CI: 0.061, 0.102)) on PF. The estimate of the causal effect using the WM method was confirmed for both ADHD and SZ, but not for BD.

The results are displayed in Fig. 3 (see also the scatterplots in Supplementary Figs. 5–13 and 23–26) and in Supplementary Table 5 (see also Supplementary Table 6 for conversion to odds ratios).

Steiger tests indicated that the causal direction between the exposure and outcome was correct in all the analyses. Cochran's *Q* heterogeneity statistics were significant in all the analyses (except for the effects of PF on ANX and P on OCD), which is suggestive of pleiotropy. Furthermore, we looked at the MR Egger regression (MR-Egger) intercept, which significantly deviated from 0 in the analysis of PF against ADHD ($P = 0.001$) and of PF against ASD ($P = 0.035$), suggesting that pleiotropy was unbalanced in these relationships. The MR-Egger causal effect, which provides a better estimate in cases of unbalanced pleiotropy than IVW MR, yielded uncertain results for the effect of PF on ADHD ($\beta = -0.232$ (95% CI: −0.568, 0.104)) and evidence of an

inverse relationship between PF and ASD ($\beta = -0.477$ (95% CI: −0.899, −0.0.54), Egger's intercept $P = 0.035$). MR pleiotropy residual sum and outlier (MR-PRESSO) did not detect bias in the estimates due to horizontal pleiotropy in all the estimates. Causal Analysis Using Summary Effect Estimates (CAUSE) confirmed the causal effect of PF on ADHD bidirectionally, and unidirectionally on AN, MDD and PTSD. No significant difference between the sharing and the causal models was found for the effect of PF on SZ bidirectionally. The results of CAUSE are presented in Supplementary Table 7.

Leave-one-out analyses, in both the forward and backward directions, showed that the direction of the effect of each SNP is consistent with the direction of the overall causal estimate. In addition, for most of the analyses, a change in the estimations of less than 10% was obtained by leaving out each SNP, suggesting that none of the genetic variants were overly influential. The results of the leave-one-out analyses are provided in Supplementary Figs. 14–22 and 27–29. Finally, to test whether our results were robust to bias due to reverse causality, we repeated the MR analyses on a subset of SNPs selected through Steiger filtering. This sensitivity analysis confirmed the results of the main analysis, except for the effect of PF on OCD, which was no longer significant after Steiger filtering ($\beta = -0.013$ (95% CI: −0.199, 0.173)). The results are presented in Supplementary Table 8.

To further explore whether the effect of poverty on mental illness is driven by specific indicators, we performed bidirectional univariable MR using each of the poverty indicators (that is, HI, OI and SD) as the exposure. The results are presented in Fig. 4; Supplementary Tables 9 and 10 (HI), 13 and 14 (OI), and 17 and 18 (SD); and Supplementary Figs. 30–107. The HI GWAS had a higher level of power than the OI and SD GWAS, as indicated by the number of SNPs selected for the analyses (~50, ~30 and ~10, respectively).

Overall, the direction of the associations obtained using the poverty common factor was consistent with those obtained for each indicator, where bidirectional causal effects of poverty on some mental illnesses were observed. Specifically, using HI as a poverty measure, we found evidence of bidirectional effects with ADHD ($\beta = -0.830$ (95% CI: −1.01, −0.647); $\beta = -0.103$ (95% CI: −0.120, −0.086); IVW forward and backward, respectively) and SZ ($\beta = -0.415$ (95% CI: −0.565, −0.265); $\beta = -0.031$ (95% CI: −0.037, −0.024); IVW forward and backward, respectively) and unidirectional causal effects on MDD ($\beta = -0.422$ (95% CI: −0.589, −0.255)) and AN ($\beta = 0.448$ (95% CI: 0.191, 0.704)). However, the effect of HI on ADHD was biased by high heterogeneity and unbalanced pleiotropy (Egger's intercept $P < 0.001$), with the MR-Egger estimate yielding inconsistent results ($\beta = 0.606$ (95% CI: −0.684, 1.90)). The bidirectional effect of OI on ADHD ($\beta = -0.859$ (95% CI: −1.05, −0.669); $\beta = -0.102$ (95% CI: −0.120, −0.083); IVW forward and backward, respectively) and unidirectional effect on AN ($\beta = -0.187$ (95% CI: −0.344, −0.030)) and MDD ($\beta = -0.322$ (95% CI: −0.502, −0.143)) were also confirmed using OI as a measure of poverty. The effect of OI on SZ was also confirmed bidirectionally ($\beta = -0.187$ (95% CI: −0.344, −0.030); $\beta = -0.010$ (95% CI: −0.017, −0.003)), but sensitivity analyses using CAUSE and Steiger filtering selection detected potential bias due to pleiotropy in the forward analysis. Notably, the effect of OI on ADHD changed after Steiger filtering, with evidence of pleiotropy (Egger's intercept, $P = 0.033$) and a non-significant MR-Egger estimate ($\beta = 0.094$ (95% CI: −1.36, 1.55)). Backward analyses using OI as the outcome confirmed the effect of BD on poverty using the IVW method but not the WM method ($\beta = 0.042$ (95% CI: 0.028, 0.057); $\beta = 0.020$ (95% CI: −0.005, 0.046); IVW and WM, respectively) as found using the common factor poverty; but new evidence was found for the causal effect of AN on OI ($\beta = 0.045$ (95% CI: 0.011, 0.080)), which was not replicated using the poverty common factor or other measures of poverty. Finally, using SD as a measure of poverty, the bidirectional effect on SZ ($\beta = 0.213$ (95% CI: 0.046, 0.380); $\beta = 0.042$ (95% CI: 0.026, 0.058); IVW forward and backward, respectively) and the unidirectional effect on AN ($\beta = -0.352$ (95% CI: −0.618, −0.087)) were confirmed. The bidirectional effect

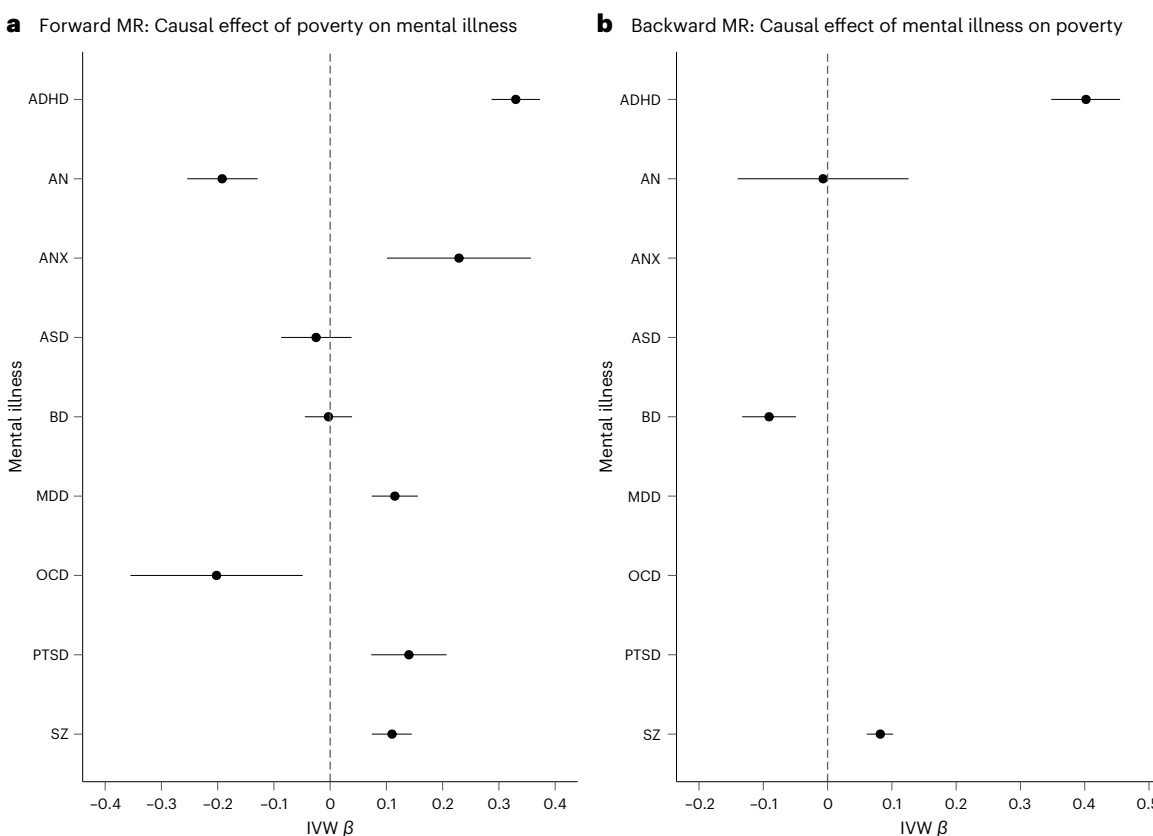

**a** Forward MR: Causal effect of poverty on mental illness

**b** Backward MR: Causal effect of mental illness on poverty

**Fig. 3 | Results of univariable bidirectional MR analysis of poverty against mental illness. a**, Forward analysis. **b**, Backward analysis. Poverty is a latent variable built using HI as the unit identification, so that an increase in the indicator's load stands for increased income; the regression coefficients have therefore been reversed to facilitate interpretation of the effect of poverty. The effect estimates on the *x* axis are log-odds for binary traits (that is, for mental illnesses) and unstandardized linear regression coefficients for continuous traits (that is, for poverty); the error bars represent 95% CIs. Poverty factor, *N* = 453,688; ADHD, *N* = 225,534; AN, *N* = 72,517; ANX, *N* = 21,761; ASD, *N* = 46,350; BD, *N* = 413,466; MDD, *N* = 138,884; OCD, *N* = 9,725; PTSD, *N* = 206,655; SZ, *N* = 320,404.

of SD on ADHD was also replicated ($\beta$ = 0.713 (95% CI: 0.504, 0.922); $\beta$ = 0.222 (95% CI: 0.181, 0.262); IVW forward and backward, respectively), with stronger evidence for the reverse effect as indicated by Steiger tests (forward, *P* = 0.079; backward, *P* < 0.001). However, when we applied Steiger filtering selection, the effect of SD on ADHD held strong ($\beta$ = 0.557 (95% CI: 0.331, 0.784)). None of the other effects of SD remained significant after Steiger filtering, which is consistent with potential bias due to reverse causation in the relationship between SD and mental illness (for the CAUSE and Steiger filtering MR results, see Supplementary Tables 11 and 12 (HI), 15 and 16 (OI), and 19 and 20 (SD)).

**Bidirectional univariable MR of HI categories against mental illness**

To explore the shape of the relationship between HI and mental health, and specifically to answer the question whether there is a particular HI threshold at which the effect of HI on mental health kicks in, we used GWAS data for the HI categories and investigated the effect of each category on the considered mental illnesses. That led to the creation of four dichotomous dummy variables: (1) low HI (LHI), for which cases were those <£18,000 and controls were those ≥£18,000; (2) low-mid HI (LMHI), for which cases were those <£29,999 and controls were those ≥£29,999; (3) mid-high HI (MHHI), for which cases were those >£52,000 and controls were those ≤£52,000; and (4) high HI (HHI), for which cases were those >£100,000 and controls were those ≤£100,000. For each of these categories, bidirectional univariable MR was performed.

As shown in Fig. 5 and Supplementary Tables 21–24 (see also Supplementary Table 25 for conversion to odds ratios), we found a significant causal effect of being in the lowest HI class on ANX ($\beta$ = 0.628 (95% CI: 0.095, 1.16)), BD ($\beta$ = 0.306 (95% CI: 0.125, 0.487)), MDD ($\beta$ = 0.351 (95% CI: 0.189, 0.513)) and PTSD ($\beta$ = 0.506 (95% CI: 0.253, 0.759)), and bidirectional effects on ADHD (forward, $\beta$ = 0.610; (95% CI: 0.415, 0.805); backward, $\beta$ = 0.210 (95% CI: 0.170, 0.250)) and SZ (forward, $\beta$ = 0.648 (95% CI: 0.488, 0.808); backward, $\beta$ = 0.082 (95% CI: 0.066, 0.098)). In the LMHI class, the direction of the associations observed in the LHI class was maintained, and the effect size decreased, consistent with our leading hypothesis of more deleterious effects of poverty on mental health at very low levels of income. The exceptions were AN, ASD and OCD, for which being in the LMHI class was protective ($\beta$ = −0.309 (95% CI: −0.526, −0.091); $\beta$ = −0.268 (95% CI: −0.488, −0.047); $\beta$ = −0.774 (95% CI: −1.34, −0.212), respectively). The analyses in the higher classes (that is, MHHI and HHI) showed that with each incremental increase or decrease of income, there is a corresponding effect on mental illness. For instance, AN and ASD (but not OCD) were causally associated with higher incomes, with stronger evidence for AN ($\beta_{\text{MHHI}\rightarrow\text{AN}}$ = 0.371 (95% CI: 0.193, 0.550); $\beta_{\text{MHHI}\rightarrow\text{ASD}}$ = 0.480 (95% CI: 0.303, 0.658); $\beta_{\text{HHI}\rightarrow\text{AN}}$ = 0.333 (95% CI: 0.065, 0.600); $\beta_{\text{HHI}\rightarrow\text{ASD}}$ = 0.094 (95% CI: −0.121, 0.308)), whereas the associations with ADHD, MDD, PTSD and SZ had the opposite sign, confirming the causal relationship with lower income. Notably, ADHD and SZ showed a consistent bidirectional pattern across income levels, where a lower income was causally related to these illnesses and the illnesses were also causal factors for lower incomes. This evidence supports a vicious cycle between poverty and severe mental illness.

The effect of the HI levels on BD showed a U-shaped distribution, in which BD was causally associated with both low and high HI

**a** Forward MR: Causal effect of poverty indicators on mental illness **b** Backward MR: Causal effect of mental illness on poverty indicators

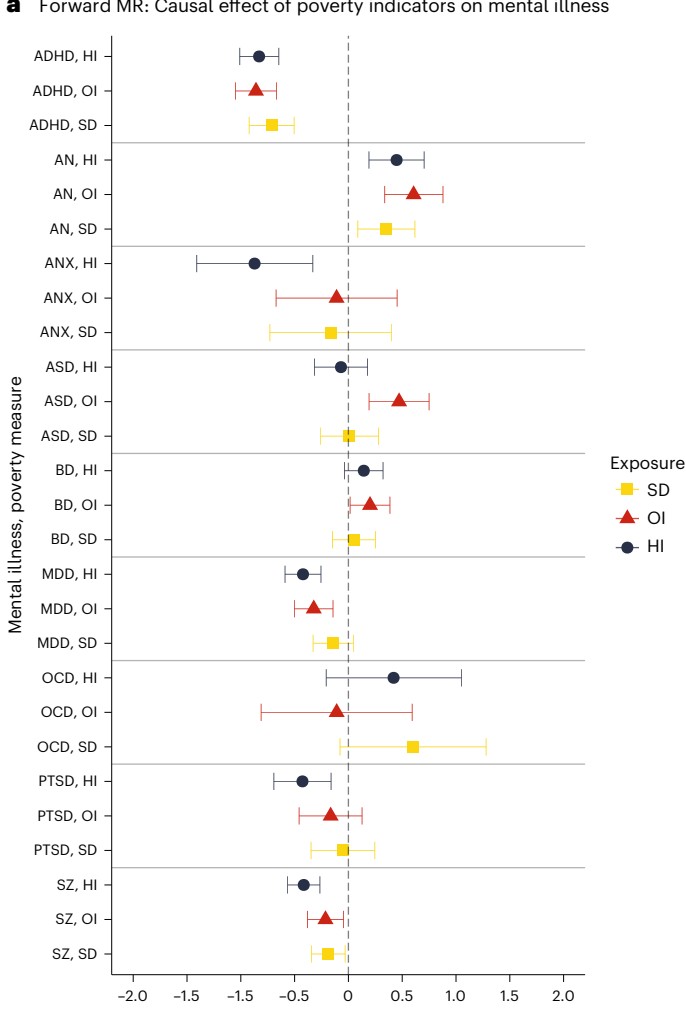

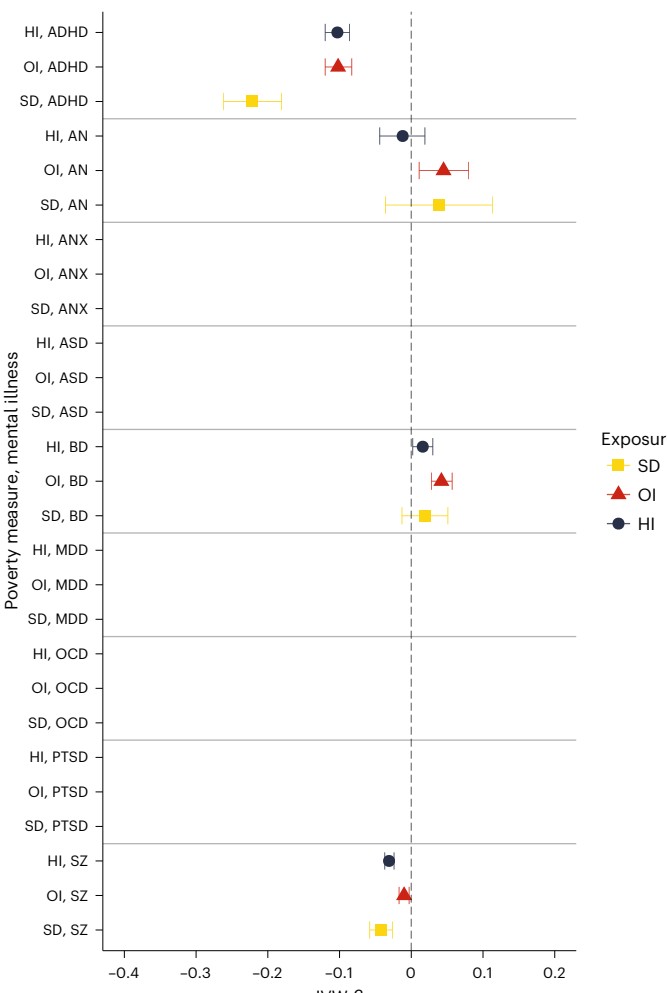

**Fig. 4 | Results of univariable bidirectional MR analysis for each poverty measure against mental illness. a**, Forward analysis. **b**, Backward analysis. To have a consistent direction of the effect, the SD effect has been reversed. The effect estimates on the *x* axis are log-odds for binary traits (that is, for mental illnesses) and unstandardized linear regression coefficients for continuous traits (that is, for the poverty indicators); the error bars represent 95% CIs. Missing results are due to an insufficient number of SNPs selected for the MR analysis. HI, *N* = 379,598; OI, *N* = 282,963; SD, *N* = 440,350; ADHD, *N* = 225,534; AN, *N* = 72,517; ANX, *N* = 21,761; ASD, *N* = 46,350; BD, *N* = 413,466; MDD, *N* = 138,884; OCD, *N* = 9,725; PTSD, *N* = 206,655; SZ, *N* = 320,404.

($\beta_{\text{LHI}\rightarrow\text{BD}}$ = 0.306 (95% CI: 0.125, 0.487); $\beta_{\text{MHHI}\rightarrow\text{BD}}$ = 0.179 (95% CI: 0.053, 0.304); $\beta_{\text{HHI}\rightarrow\text{BD}}$ = 0.608 (95% CI: 0.423, 0.792)). However, these associations were not replicated using the WM method, and MR-PRESSO detected significant distortion in the estimate due to pleiotropy in the relationship between BD and MHHI, with non-significant outlier-adjusted causal estimates ($\beta$ = 0.002 (95% CI: −0.039, 0.042)).

The estimate of the causal effect using the WM method was consistent in magnitude for the bidirectional effects of LHI on ADHD and SZ and for the unidirectional effects of LHI on ANX, MDD and PTSD and of MHHI and HHI on ASD.

*Q* statistics were suggestive of high levels of heterogeneity in most of the analyses; however, MR-Egger showed a significant intercept and causal effect only for the relationship between MHHI and BD ($\beta$ = 1.69 (95% CI: 0.376; 3.00); Egger's intercept *P* = 0.033). Steiger tests indicated that the causal direction between the exposure and outcome was correct in all the analyses except those on ASD (*P* = 0.994 and *P* = 0.057, respectively, for the associations with LMHI and MHHI).

The MR analyses on a subset of SNPs selected through Steiger filtering confirmed a stronger causal effect of the lowest HI level on ADHD, MDD, PTSD and SZ. The protective effect of MHHI on ASD was also confirmed. Interestingly, the effects observed in the main analysis

were not confirmed for BD and AN, yielding evidence of reverse causation rather than direct causality for these associations. These results can be found in Supplementary Table 26.

## Multivariable MR of poverty factor and CA against mental illness

Like other measures of SES, genetic effects are unlikely to influence poverty directly. Rather, genetic effects are likely to influence poverty through intermediary traits (such as health, personality, intelligence and other characteristics) that are themselves heritable[19]. Previous studies have shown that CA (referred to as general cognitive function, performance or intelligence) is one likely causal factor in income differences[19] as well as being genetically associated with mental illness[20]. We used multivariable MR (MVMR) to estimate the effect of poverty on each mental illness, while controlling for CA. First, we performed bidirectional univariable MR of CA on mental illness, finding evidence supporting a bidirectional inverse causal relationship between CA and ADHD ($\beta$ = −0.638 (95% CI: −0.720, −0.556); $\beta$ = −0.151 (95% CI: −0.172, −0.131); forward and backward direction, respectively) and SZ ($\beta$ = −0.297 (95% CI: −0.365, −0.228); $\beta$ = −0.055 (95% CI: −0.063, −0.047); forward and backward direction, respectively), a unidirectional negative causal

**a** Forward MR: Causal effect of household income levels on mental illness

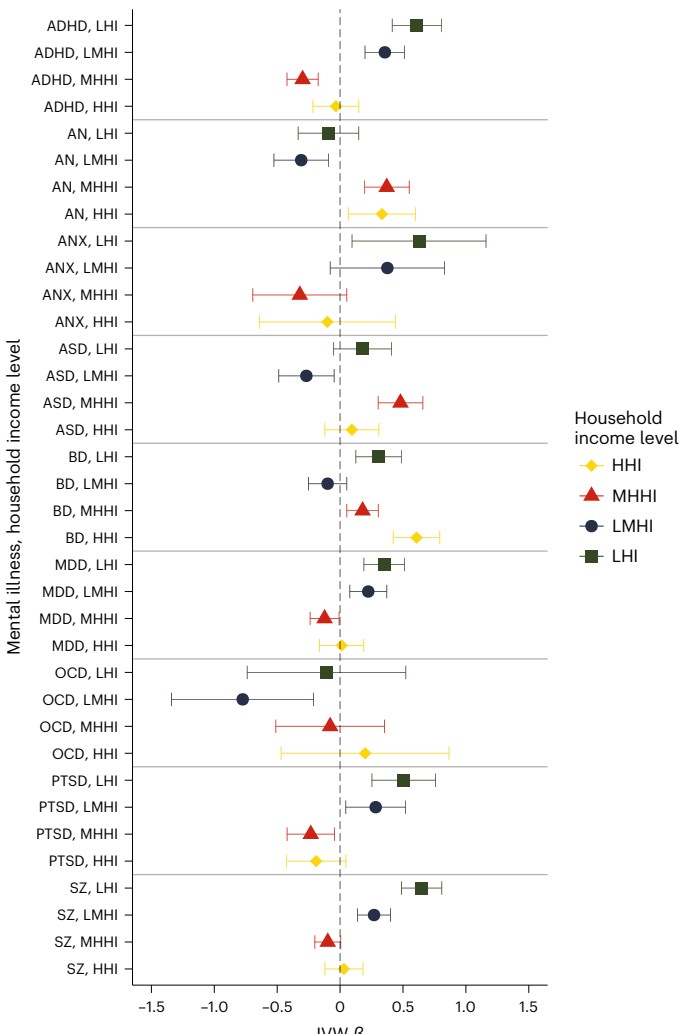

**b** Backward MR: Causal effect of mental illness on household income levels

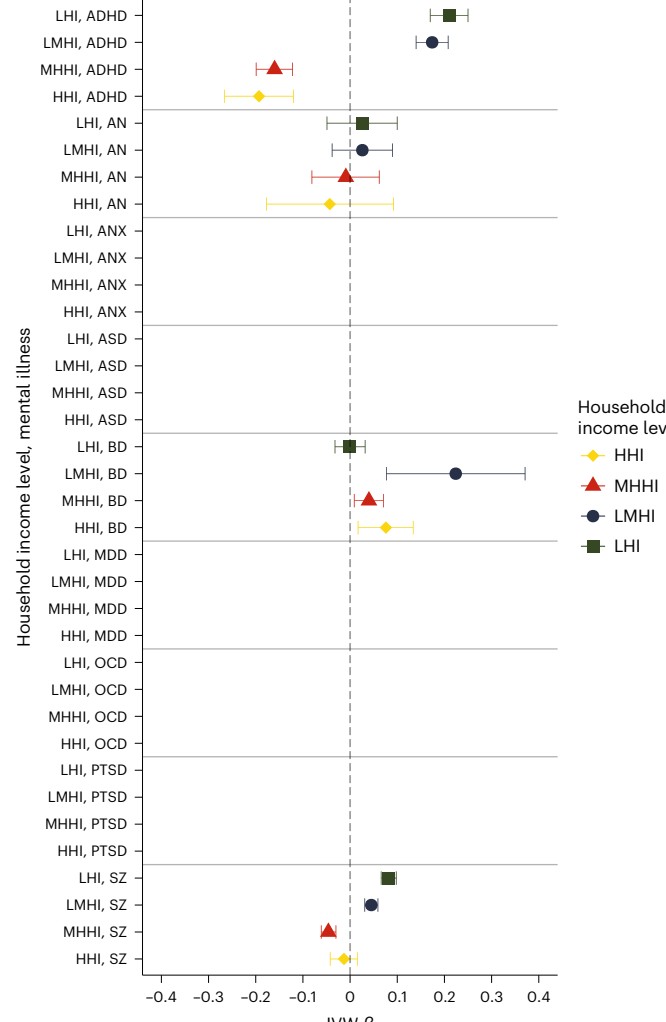

**Fig. 5 | Results of univariable bidirectional MR analysis of HI levels against mental illnesses. a**, Forward analysis. **b**, Backward analysis. For LHI, cases <£18,000 and controls ≥£18,000; for LMHI, cases <£29,999 and controls ≥£29,999; for MHHI, cases >£52,000 and controls ≤£52,000; and for HHI, cases >£100,000 and controls ≤£100,000. The effect estimates on the *x* axis are log-odds for both the forward and backward analyses given that all traits are binary; the error bars represent 95% CIs. Missing results are due to an insufficient number of SNPs selected for the MR analysis. HI, *N* = 379,598; ADHD, *N* = 225,534; AN, *N* = 72,517; ANX, *N* = 21,761; ASD, *N* = 46,350; BD, *N* = 413,466; MDD, *N* = 138,884; OCD, *N* = 9,725; PTSD, *N* = 206,655; SZ, *N* = 320,404.

effect of CA on MDD and PTSD ($\beta = -0.140$ (95% CI: −0.215, −0.065) and $\beta = -0.139$ (95% CI: −0.264, −0.013), respectively) and a unidirectional causal effect of CA on AN and ASD ($\beta = 0.306$ (95% CI: 0.190, 0.422) and $\beta = 0.310$ (95% CI: 0.193, 0.427), respectively). These results are displayed in Supplementary Tables 27–30 and Supplementary Figs. 108–133. The MVMR analysis yielded results supporting a causal effect of PF on ADHD, AN, ANX, MDD, OCD, PTSD and SZ, beyond CA. Still, the effect of PF on mental illness decreased when including CA in the model. These results suggest that the univariable MR results for PF on mental illness are slightly biased by CA, but a direct effect of PF on ADHD, AN and ANX remains, nonetheless. The results of MVMR are shown in Table 1.

Importantly, and unlike the univariable models, the inclusion of CA in the MVMR model produced highly dissimilar results comparing between the poverty factor and each indicator of poverty. First, the inclusion of CA in the MVMR model removed the effect of HI on mental illness. Second, for both OI and SD, some significant effects remained, where OI demonstrated a direct effect on ASD and SD showed a direct effect on AN. The results of these additional MVMR models are shown in

Supplementary Tables 30–33. Taken together, these results support the idea that the majority of the genetic effects that link poverty to mental illness do act on CA. However, by using a general factor of poverty, the resulting increase in statistical power facilitated the discovery of genetic effects linking poverty to mental health independent of the effects of CA. Therefore, while CA is correlated with poverty, it does not fully account for the observed genetic effects on mental health. These findings, when considered alongside the results of the MR of PF and CA, point to CA fitting as an upstream component in the complex causal pathway linking poverty to mental illness.

## Discussion

Building on data from 18 GWAS, this study provides support for a causal relationship between poverty and some mental illnesses. Jointly modelling different indicators of poverty using genomic structural equation modelling and subsequent MR provided converging evidence supporting bidirectional causal effects of poverty and ADHD/SZ, a unidirectional causal effect of poverty on MDD and an inverse causal relationship between poverty and AN. Notably, the evidence

**Table 1 | MVMR results for poverty and CA on mental illness**

| MVMR | No. of SNPs | IVW β (95% CI) | IVW P |
|---|---|---|---|
| Outcome: ADHD | 203 | | |
| Exposure 1: PF | | 0.232 (0.169, 0.295) | **7.28×10⁻¹³** |
| Exposure 2: CA | | −0.477 (−0.612, −0.342) | **4.31×10⁻¹²** |
| Outcome: AN | 205 | | |
| Exposure 1: PF | | −0.109 (−0.183, −0.035) | **0.004** |
| Exposure 2: CA | | 0.149 (−0.008, 0.306) | 0.062 |
| Outcome: ANX | 207 | | |
| Exposure 1: PF | | 0.114 (0.007, 0.221) | **0.036** |
| Exposure 2: CA | | −0.344 (−0.571, −0.117) | **0.003** |
| Outcome: ASD | 208 | | |
| Exposure 1: PF | | 0.022 (−0.054, 0.099) | 0.564 |
| Exposure 2: CA | | 0.217 (0.054, 0.379) | **0.009** |
| Outcome: BD | 208 | | |
| Exposure 1: PF | | 0.020 (−0.044, 0.085) | 0.538 |
| Exposure 2: CA | | 0.006 (−0.132, 0.144) | 0.931 |
| Outcome: MDD | 208 | | |
| Exposure 1: PF | | 0.076 (0.030, 0.123) | **0.001** |
| Exposure 2: CA | | −0.091 (−0.190, 0.008) | 0.071 |
| Outcome: OCD | 208 | | |
| Exposure 1: PF | | −0.202 (−0.340, −0.0064) | **0.004** |
| Exposure 2: CA | | 0.320 (0.024, 0.615) | **0.034** |
| Outcome: PTSD | 208 | | |
| Exposure 1: PF | | 0.111 (0.048, 0.173) | **4.90×10⁻⁴** |
| Exposure 2: CA | | −0.160 (−0.294, −0.027) | **0.019** |
| Outcome: SZ | 208 | | |
| Exposure 1: PF | | 0.092 (0.018, 0.167) | **0.015** |
| Exposure 2: CA | | −0.297 (−0.457, −0.136) | **2.94×10⁻⁴** |

'IVW' indicates multivariable MR via the IVW method (random effects); the β effect estimates are log-odds. Poverty is a latent variable built using HI as a unit identification, so that an increase in the indicator's load stands for increased income; the regression coefficients have therefore been reversed to facilitate interpretation of the effect of poverty. All statistical tests are two-sided. The P values were not adjusted for multiple testing; therefore, P<0.05 was considered significant (and is reported in bold).

for a causal relationship between poverty and ADHD was influenced by unbalanced pleiotropy. Our findings, in two Western-ancestry samples, complement the available evidence on social inequalities in mental health collected across different countries, samples and study designs[2,7,11,21,22], including previous MR studies focusing on MDD, HI and unemployment[23–25]. However, our research adds evidence that the relationship between poverty and mental illness is valid across different measures of poverty and extends the investigation to a wider range of mental illnesses.

This study primarily builds on genetic evidence and therefore warrants caution with respect to the conceptualization of poverty and mental illness. Moreover, while poverty and other measures of SES are heritable traits, it is very unlikely that there are direct genetic effects. Genetic associations with socio-economic traits are probably the result of vertical pleiotropy. Vertical pleiotropy describes instances where one trait is causally associated with a second trait. In these cases, the genetic effects that act on the first trait will also be detected as being associated with the second[26]. In that context, MR can best be viewed as a way to approximately randomly assign heritable traits that give rise to income differences. Two key candidates for such underlying

traits are intelligence and personality. For example, intelligence may lead to both educational advantage and socio-economic success as well as healthier behaviours and thus lead to good mental health[27–31]. In this study, we adjusted for CA and made two additional observations. First, when we controlled for CA, the causal effect of the general factor of poverty decreased by around 30% compared with univariate estimates, indicating that CA was a contributor to the causal association between poverty and mental illness. Second, when we examined each of the indicators of poverty separately, the effect of HI on mental illness was no longer statistically significant after controlling for CA. For both OI and SD, most of the evidence of a direct causal effect of poverty was absent after adjusting for CA; however, lower levels of OI remained causally associated with increases in ADHD, and higher SD remained in an inverse association with AN. Nevertheless, the higher power obtained by using the common factor of poverty rather than the individual indicators probably facilitated the detection of the influence of heritable poverty traits above and beyond the role of CA and enabled the detection of the causal role of poverty in mental health.

Of note is the vast literature showing that health characteristics (for example, obesity), physical appearance and mental state can influence economic outcomes, including employment opportunities and wages[32–34]. Our results showing a causal effect of poverty on mental illness therefore should not be taken as proof supporting genetic determinism, but rather as epidemiological evidence of the detrimental effects of poverty and socio-economic inequalities on mental health, regardless of the mechanisms (see also the Frequently Asked Questions in the Supplementary Information).

The causal relationship between poverty and mental illness is likely to involve material, psychological, behavioural and biological pathways. For example, those with heritable traits linked to lower levels of poverty may be more able to move away from stressful environments/jobs/neighbourhoods and so place themselves at a lower risk of disorders[35]. Furthermore, the level of development of the welfare system may be a material mediator of the health-damaging effects of income losses[36,37]. Psychosocial mechanisms are a result of the interaction between people's social environment and their feelings: living on a low income is stressful, and at the same time people in disadvantaged situations may have fewer resources to cope with difficult circumstances. Increasingly, biological research is providing evidence showing how experiences such as social defeat can 'get under the skin', causing biochemical changes in the body and brain and increasing the risk of developing mental health problems[38,39]. Other contributing factors to the relation between poverty and mental illness are the negative health behaviours that are more prevalent in socially disadvantaged groups[40], probably as a consequence of the higher financial cost of healthy behaviours. For example, a healthy diet is more expensive than processed foods, and joining a gym or sporting clubs can be costly. Moreover, unhealthy behaviours such as smoking and drinking alcohol may be used as coping strategies for stressful situations[41].

The findings of this study reiterate the need to further unravel the roles of poverty and mental illness and to use this insight to advance mental health for all. The choice of which action to take to address the problem is a political matter, but attention is warranted considering increasing income inequalities worldwide[42], as well as the increasing incidence of mental illness[43,44].

Previous studies examining antipoverty programmes reported positive and sustained effects on mental health[45], including reduction of depression and suicide rates[11,46,47]. For example, a recent randomized controlled trial found that cash incentives provided to low-income individuals led to meaningful improvements in depression[48]. Additionally, specific economic interventions such as family assets, employment support and rental assistance may be effective in enhancing the mental health of programme beneficiaries[49,50]. However, it is important to note that previous quasi-experimental research using natural experiments (such as on the lottery winner population) have provided

supportive evidence for a small association between SES and mental health[51], and qualitative studies have shown that receiving money could be perceived as a debt or burdensome responsibility[52,53], which may contribute to increased stress[54,55]. Further research is therefore necessary to identify the factors within economic support interventions (for example, microcredit, loans and personal health budgets) that influence mental health outcomes. Early studies have found that that combining antipoverty interventions, such as financial incentives and financial mentoring programmes, appears more effective in improving mental health than providing financial incentives alone[47]. Given these complexities, policymakers could consider policies that can be beneficial for both individuals living in poverty and those living with severe mental illness. Policymakers should also consider a key finding of this study about the effect of CA in the causal pathway linking poverty to mental illness. CA is closely associated with educational attainment and occupational status, which are often regarded as SES variables relevant to health[27,29,56]. Although the genetic relationship between CA and mental illness differs from the genetic relationship between education and mental illness[20,57], educational attainment is a likely causal factor in CA differences[58-60] and is a more straightforward target for interventions than CA. Future research should thus explore strategies aimed at facilitating individuals' participation in education, which may lead to better mental health[8,61].

This study has limitations that should be considered when interpreting the results. First, for ANX, ASD, MDD, OCD and PTSD, the GWAS sample sizes were relatively small, precluding their use as exposure variables for investigating the impact of these disorders on poverty. This is due to the insufficient number of SNPs retained following the selection, clumping and harmonization steps. For SNP selection, we used the threshold of $P < 5 \times 10^{-8}$, which is recommended by MR guidelines to ensure the validity of the MR relevance assumption[62]. We could have relaxed this threshold to include additional SNPs, but this could introduce bias. We therefore refrained from performing those MR analyses and trust that future, more powerful GWAS on these conditions will yield more suitable instruments for MR, bridging this gap. Second, most genetic studies were conducted in populations of European ancestry from high-income countries. Generalizability to populations from other ancestries and from low- and middle-income countries is therefore uncertain[63]. In particular, this raises questions about the specificity and universality of the effects of poverty on mental health in different cultures and contexts with different political and social systems. Third, it is well acknowledged that psychiatric phenotypes are complex and heterogeneous, which generally translates to low power. This is reflected by the relatively modest effects of each poverty indicator on mental illness. Fourth, it is essential to acknowledge the potential influence of the dynastic effect, wherein characteristics transmitted across generations (such as the association between a parent's genotype and the offspring's phenotype) are a known source of bias in MR studies as they violate the second assumption (that is, independence)[64]. Detecting the exact magnitude of bias resulting from this effect is challenging, as current sample sizes preclude such analyses today. Future studies examining the causal link between poverty and mental health outcomes in an MR framework will be better placed to assess any influence of dynastic effects on the causal estimates identified. We also advise triangulation of our findings using complementary research methods in future studies. Fifth, it should be noted that previous research pointed out that UK Biobank participants are on average healthier and live in less socio-economically deprived areas than the UK average[65]. This could introduce potential selection bias in MR studies using UK Biobank data, inflating the risk of type I error[66,67]. However, the impact of selection bias is particularly strong when the selection effect is large—for instance, when studying disease progression, secondary diseases or specific subpopulations, such as elderly people[66]. In the context of our research, the selection effect is likely to be low. Furthermore, there is evidence that the UK Biobank is

### Table 2 | GWAS information for each phenotype

| Trait | N | No. of cases | Consortium | Reference |
|---|---|---|---|---|
| ADHD | 225,534 | 38,691 | PGC | Demontis et al.[79] |
| AN | 72,517 | 16,992 | PGC | Watson et al.[80] |
| ANX | 21,761 | 7,016 | PGC | Otowa et al.[81] |
| ASD | 46,350 | 18,381 | PGC | Grove et al.[82] |
| BD | 413,466 | 41,917 | PGC | Mullins et al.[83] |
| CA | 248,482 | NA | UKB | Hill et al.[20] |
| HI | 379,598 | NA | UKB | NA |
| MDD | 138,884 | 43,204 | PGC | Howard et al.[84] |
| OCD | 9,725 | 2,688 | PGC | Arnold et al.[85] |
| OI | 282,963 | NA | NA | Kweon et al.[76] |
| PTSD | 206,655 | 32,428 | PGC | Nievergelt et al.[86] |
| SD | 440,350 | NA | UKB | NA |
| SZ | 320,404 | 76,755 | PGC | Trubetskoy et al.[87] |

The traits are presented in alphabetical order. NA, not applicable; PGC, Psychiatric Genomic Consortium; UKB, UK Biobank.

sufficiently large and heterogeneous to provide valid scientific inferences of associations between exposures and health conditions, which is in line with the aim of our research[65]. Sixth, the selection of SNPs associated with the latent poverty factor was performed in the UK Biobank, leading to the potential risk of overestimation of the SNP–phenotype association due to winner's curse bias. However, the impact of this bias is generally not substantial in large samples, such as the UK Biobank[68]. Other limitations of MR concern temporality and linearity: MR is thought to provide estimates of lifetime risk and assumes linear effects. Although the use of MR on HI levels and the inclusion of mental illness with onsets across the whole person's lifespan (for example, ranging from ADHD and ASD to MDD and ANX) may have mitigated these limitations, future studies should particularly focus on assessing whether there are critical windows or acute reactions to poverty exposure. Finally, the genetic variants captured by our measure of poverty are likely to have pleiotropic effects[69]. However, to break the assumptions of MR, it is not sufficient for the genetic variants in the instrumental variable to have pleiotropic effects[70]. The genetic variants used as instrumental variables must have horizontally pleiotropic effects mediated via mechanisms other than those captured by poverty. Should genetic variants have vertically pleiotropic effects (for example, SNP → neuron → intelligence → education → poverty → health → psychiatric disorder), then our MR-derived causal estimates will not be biased. Furthermore, if the SNPs affect other phenotypes, but these phenotypes do not affect outcome, then these effects will not bias our MR estimates. It is possible that the genetic variants identified in our poverty GWAS do have horizontally pleiotropic effects, but it is unclear what mechanisms would mediate such effects (for example, personality). In the current study, we investigated potentially pleiotropic effects using MVMR to examine the role of CA[71]. Future research should use MVMR to investigate the roles of other traits that link poverty to mental health outcomes.

## Methods
### Study design and data sources
We conducted a two-sample MR study using summary-level GWAS data that for the most part were publicly available. Specifically, for this study, new GWAS on HI and SD were performed using UK Biobank data under application number 10279 (see 'Data availability' for the links to these data), the OI data are publicly available and can be downloaded from https://osf.io/rg8sh/, the CA data were obtained from the author of the relevant publication[20], and the summary statistics files for the

mental illness phenotypes are publicly available for download from the Psychiatric Genomic Consortium (PGC) website at https://pgc.unc.edu/for-researchers/download-results/. Further information on the sources of the summary data is reported in Table 2. Ethical approval was obtained in all original studies.

MR relies on three main assumptions on the validity of the genetic instrument variable: relevance, independence and exclusion restriction[72]. These require that the genetic variant (1) is related to the exposure (that is, only SNPs that reach genome-wide significance in the association with the exposure, or $P < 5 \times 10^{-8}$, are used), (2) is not correlated with confounders in the exposure–outcome relationship and (3) affects the outcome only through the risk factor (that is, the 'no horizontal pleiotropy' rule).

### Poverty and CA instruments

For this research, seven new GWAS were performed: five on HI, one on SD and one on the common factor of poverty. The GWAS on CA and OI were publicly available. A total of 379,598 participants of European ancestry provided genotype data and data on their level of yearly HI before tax. HI was primarily analysed as a continuous variable. To investigate whether the relation between HI and mental illness is particularly strong at any specific level of income, we further categorized HI as a dichotomous variable, defining the cases in the following way: (1) LHI: less than £18,000; (2) LMHI: less than £29,999; (3) MHHI: more than £52,000; (4) HHI: more than £100,000. One GWAS for each HI category versus all the other categories was performed. A total of 440,350 individuals of European ancestry had genotype data and data on their level of SD, measured with the Townsend Deprivation Index, which was analysed as a continuous variable. GWAS of HI and SD were performed in REGENIE (version 3.1.3)[73]; for detailed methods, see the Supplementary Information. Previously published GWAS summary data on CA of 248,482 individuals were included[20]. The CA measure was derived from an already existing dataset of multiple cohorts in which each participant had been administered a battery of cognitive tests[20]. Correlations between tests of CA are high, with estimates between $r = 0.8$ and $r = 1.0$ being reported[74,75]. Further information on the CA data can be found in the Supplementary Information. In this study, CA was analysed as a continuous variable.

The results from a meta-analysis of two OI GWAS[76], including the UK Biobank and the Health and Retirement Study (the latter involving participants from white American ancestry), were used to generate the instruments for OI, which were analysed as continuous variables.

We estimated a latent factor GWAS of poverty (PF) by jointly modelling cross-trait liability of continuous HI, OI and SD, using a common factor model in genomic structural equation modelling[77]. Before running the multivariable GWAS, we aligned the alleles across HI, SD and OI traits using the HAPMAP3 reference panel, and summary statistics passed quality control by selecting SNPs with minor allele frequency >0.01 and INFO >0.9. We then estimated the LDSC[78] across HI, OI and SD using the 1000G European reference panel. Finally, we combined the LDSC output with the HI, OI and SD summary statistics to run the multivariable common factor GWAS. This common factor poverty GWAS was built using HI as the unit of identification, and the effective sample size was 453,689. To allow the estimation of poverty instead of income in the MR analyses on mental illness, we reversed the regression coefficients. The full details on the genomic structural equation modelling methods are available in the Supplementary Information.

### Mental illness instruments

Results from the most updated GWAS of ADHD[79], AN[80], ANX[81], ASD[82], BD[83], MDD[84], OCD[85], PTSD[86] and SZ[87] were obtained. The two-sample MR method requires minimal sample overlap between the exposure and outcome GWAS[88]; therefore, mental illness summary-level data were obtained mainly from the PGC, excluding UK Biobank participants. The GWAS of the mental illness traits considered were provided and analysed as case/control. The effective sample size across the cohorts

contributing to the GWAS meta-analysis was calculated for each trait and ranged from 320,000 (for SZ) to 10,000 (for OCD). Table 2 summarizes the GWAS information.

### MR

The full set of GWAS summary statistics for each exposure was first restricted to the variants reaching the genome-wide significance threshold (that is, $P < 5 \times 10^{-8}$). Then, to ensure independence between instruments, we applied a strict clumping procedure (LD $r^2 < 0.001$ within 10 Mb, using 1000G EUR as the reference panel). Following that, SNP alleles were harmonized between exposure GWAS and outcome GWAS, using action = 2 (which tries to infer positive strand alleles, using allele frequencies for palindromes), before running MR.

We conducted bidirectional univariable MR between each poverty measure and mental illness. The IVW method was used to estimate effects in the primary analyses[89], and the results are presented using forest plots. The WM[90], MR-Egger[91] and MR-PRESSO[92] methods were used as sensitivity analyses, because each method makes different assumptions regarding instrument validity. Specifically, MR-Egger and MR-PRESSO are better than IVW in cases of horizontal pleiotropy (that is, violation of the exclusion restriction assumption). In addition, we used CAUSE[93] to account for the presence of correlated or uncorrelated pleiotropy (that is, violation of the independence and exclusion restriction assumptions). Further information on MR-PRESSO and CAUSE is available in the Supplementary Information.

We present the results of the MR analyses as $\beta$ values corresponding to the log-odds for binary traits (for example, mental illness) or to the unstandardized linear regression coefficients for continuous traits (for example, poverty measures) with their respective 95% CIs. When the outcome of the analysis was binary (that is, for mental illness and HI levels), we also provide a conversion to odds ratios with corresponding 95% CIs.

We also conducted leave-one-out analyses to investigate whether the effects are driven by one or a subset of the SNPs and investigated whether the instruments represent the correct causal direction using Steiger analyses, including Steiger tests and Steiger filtering[94]. Steiger filtering is particularly useful to avoid false positive findings due to reverse causation (that is, violation of the exclusion restriction assumption). The mr_wrapper() function of the TwosampleMR package performs Steiger tests on each SNP, evaluating whether the $R^2$ of the exposure is greater than the $R^2$ of the outcome and indicating the correct direction of the association. Subsequent Steiger filtering excluded SNPs with false results, allowing MR analyses to be performed on the subset of SNPs with verified associations.

Finally, we hypothesized that CA is probably involved in the poverty and mental illness causal pathway, and our instruments for poverty are possibly related to CA. We therefore used MVMR[72] to estimate the direct effect of poverty on mental illness, independent of CA. For this purpose, we clumped the full list of SNPs from each poverty indicator GWAS (to ensure that only independent SNPs are included) and restricted it to those SNPs found in the outcome GWAS, and then ran the analyses on each instrument–mental illness set.

Instrument strength was quantified using the mean $F$ statistic within the univariable IVW analyses, considering a value of $F < 20$ as indicative of weak instruments[95].

All the analyses were conducted in R (version 4.4.0)[96], using the packages TwosampleMR (version 0.6.2)[97], CAUSE (version 1.2.0)[93], and GenomicSEM (version 0.0.5)[77]. The statistical tests were two-tailed. The significance threshold was $P < 0.05$. Since only one relationship was tested—that between poverty and mental illness—we did not apply a multiple-testing $P$-value correction.

### Reporting summary

Further information on research design is available in the Nature Portfolio Reporting Summary linked to this article.

## Data availability

The UK Biobank data used in this study are available via the UK Biobank data access process (http://www.ukbiobank.ac.uk/register-apply/). The PGC data can be publicly accessed via https://pgc.unc.edu/for-researchers/download-results/. The OI data are publicly available and can be downloaded from https://osf.io/rg8sh/. The CA data were obtained from the author of the relevant publication, accessible at https://www.nature.com/articles/s41380-017-0001-5. The full GWAS summary statistics on HI, SD and poverty have been uploaded to the GWAS catalogue (https://www.ebi.ac.uk/gwas/deposition/submission/6572ff9a53c4ef000109d322; study accession IDs: GCST90302879, GCST90302880, GCST90302881, GCST90302882, GCST90302883, GCST90302884 and GCST90302885).

## Code availability

The code for replicating the analyses reported in this Article can be accessed at https://github.com/MattiaMarchi/Common-factor-GWAS–MR.

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

## Acknowledgements

The authors received no specific funding for this work. We thank M. C. Murari from the University of Modena and Reggio Emilia for help with High Performance Computing Cluster. W.D.H. and C.X. are supported by a Career Development Award from the Medical Research Council (no. MR/T030852/1) for the project titled 'From genetic sequence to phenotypic consequence: genetic and environmental links between cognitive ability, socioeconomic position, and health'.

## Author contributions

Conceptualization: M.M., M.P.M.B. and W.D.H. Methodology: M.M., M.P.M.B., W.D.H., C.X. and C.H.L.T. Software: M.M., C.X., W.D.H. and C.H.L.T. Formal Analysis: M.M., W.D.H. and C.X. Resources: M.M., W.D.H., C.X., C.H.L.T., H.K. and M.P.M.B. Data Curation: M.M., M.P.M.B., W.D.H., C.X. and H.K. Writing - Original Draft: M.M., A.A., W.S. and L.-Y.C. Writing - Review & Editing: M.P.M.B., G.M.G., S.F., L.P., S.E.-L., C.H.L.T., C.X., W.D.H., M.M., A.A. and W.S. Visualization: M.M. and W.D.H. Supervision: M.P.M.B. and W.D.H. All the authors approved the final version of the manuscript.

## Competing interests

The authors declare no competing interests.

## Additional information

**Correspondence and requests for materials** should be addressed to Gian M. Galeazzi or Marco P. Boks.

[1]Department of Biomedical, Metabolic and Neural Sciences, University of Modena and Reggio Emilia, Modena, Italy. [2]Department of Mental Health and Addiction Services, Azienda USL-IRCCS di Reggio Emilia, Reggio Emilia, Italy. [3]Department of Psychiatry, Brain Center University Medical Center Utrecht, University Utrecht, Utrecht, the Netherlands. [4]Lothian Birth Cohort Studies, University of Edinburgh, Edinburgh, UK. [5]Department of Psychology, University of Edinburgh, Edinburgh, UK. [6]Department of Epidemiology, University of Groningen, University Medical Centre Groningen, Groningen, the Netherlands. [7]Department of Economics, School of Business and Economics, Vrije Universiteit Amsterdam, HV Amsterdam, the Netherlands. [8]Care Policy and Evaluation Centre, London School of Economics and Political Science, London, UK. [9]Dimence Institute for Specialized Mental Health Care, Dimence Group, Deventer, The Netherlands. [10]Present address: Department of Population Health Sciences, Institute for Risk Assessment Sciences, Utrecht University, Utrecht, the Netherlands. [11]Present address: Department of Mental Health and Addiction Services, Azienda USL-IRCCS di Reggio Emilia, Reggio Emilia, Italy. [12]Present address: Department of Psychiatry, Amsterdam UMC, Amsterdam, The Netherlands.
✉e-mail: GianMaria.Galeazzi@ausl.re.it; m.p.m.boks@umcutrecht.nl

# Reporting Summary

## Statistics

For all statistical analyses, confirm that the following items are present in the figure legend, table legend, main text, or Methods section.

| n/a | Confirmed | |
|---|---|---|
| ☐ | ☒ | The exact sample size (*n*) for each experimental group/condition, given as a discrete number and unit of measurement |
| ☐ | ☒ | A statement on whether measurements were taken from distinct samples or whether the same sample was measured repeatedly |
| ☐ | ☒ | The statistical test(s) used AND whether they are one- or two-sided *Only common tests should be described solely by name; describe more complex techniques in the Methods section.* |
| ☐ | ☒ | A description of all covariates tested |
| ☐ | ☒ | A description of any assumptions or corrections, such as tests of normality and adjustment for multiple comparisons |
| ☐ | ☒ | A full description of the statistical parameters including central tendency (e.g. means) or other basic estimates (e.g. regression coefficient) AND variation (e.g. standard deviation) or associated estimates of uncertainty (e.g. confidence intervals) |
| ☐ | ☒ | For null hypothesis testing, the test statistic (e.g. $F$, $t$, $r$) with confidence intervals, effect sizes, degrees of freedom and $P$ value noted *Give P values as exact values whenever suitable.* |
| ☒ | ☐ | For Bayesian analysis, information on the choice of priors and Markov chain Monte Carlo settings |
| ☐ | ☒ | For hierarchical and complex designs, identification of the appropriate level for tests and full reporting of outcomes |
| ☐ | ☒ | Estimates of effect sizes (e.g. Cohen's *d*, Pearson's *r*), indicating how they were calculated |

*Our web collection on statistics for biologists contains articles on many of the points above.*

## Software and code

Policy information about availability of computer code

| Data collection | No software was used. |
|---|---|
| Data analysis | The GWAS of household income and social deprivation were performed in REGENIE version 3.1.3 (1). All the other analyses were conducted in R version 4.4.0 (2), using the packages TwosampleMR version 0.6.2 (3), CAUSE version 1.2.0 (4), and GenomicSEM version 0.0.5 (5). <br><br> 1: Mbatchou, J. et al. Computationally efficient whole-genome regression for quantitative and binary traits. Nature Genetics 2021 53:7 53, 1097–1103 (2021). <br> 2: RStudio Team. RStudio: Integrated Development Environment for R. Preprint at http://www.rstudio.com/ (2022). <br> 3: Hemani, G. et al. The MR-Base platform supports systematic causal inference across the human phenome. Elife 7, (2018). <br> 4: Morrison, J., Knoblauch, N., Marcus, J. H., Stephens, M. & He, X. Mendelian randomization accounting for correlated and uncorrelated pleiotropic effects using genome-wide summary statistics. Nature Genetics 2020 52:7 52, 740–747 (2020). <br> 5: Grotzinger, A. D. et al. Genomic SEM Provides Insights into the Multivariate Genetic Architecture of Complex Traits. Nat Hum Behav 3, 513 (2019). <br><br> The custom codes used to produce the results reported in the manuscript are available at: https://github.com/MattiaMarchi/Common-factor-GWAS---MR |

For manuscripts utilizing custom algorithms or software that are central to the research but not yet described in published literature, software must be made available to editors and reviewers. We strongly encourage code deposition in a community repository (e.g. GitHub). See the Nature Portfolio guidelines for submitting code & software for further information.

## Data

Policy information about availability of data

All manuscripts must include a data availability statement. This statement should provide the following information, where applicable:

- Accession codes, unique identifiers, or web links for publicly available datasets
- A description of any restrictions on data availability
- For clinical datasets or third party data, please ensure that the statement adheres to our policy

UK Biobank data used in this study are available via the UK Biobank data access process (see http://www.ukbiobank.ac.uk/register-apply/). PGC data can be publicly accessed from https://pgc.unc.edu/for-researchers/download-results/. OI data are publicly available and can be downloaded from https://osf.io/rg8sh/. CA data were obtained from the author of the relevant publication accessible at https://www.nature.com/articles/s41380-017-0001-5. The full GWAS summary statistics of HI, SD, and poverty are uploaded on GWAS catalog (https://www.ebi.ac.uk/gwas/deposition/submission/6572ff9a53c4ef000109d322; Study accession IDs: GCST90302879; GCST90302880; GCST90302881; GCST90302882; GCST90302883; GCST90302884; GCST90302885).

## Research involving human participants, their data, or biological material

Policy information about studies with human participants or human data. See also policy information about sex, gender (identity/presentation), and sexual orientation and race, ethnicity and racism.

| Reporting on sex and gender | Our study employs Mendelian randomization, utilizing summary statistics from previously conducted GWAS to examine the causal effect of an instrumental variable on a potential outcome. It is important to note that no participants were directly enrolled, and human data was not handled in this research. Our findings have broad applicability across all sexes and genders. Although sex and gender were not directly collected or utilized in this specific Mendelian randomization study, it is worth mentioning that the majority of the GWAS studies we utilized did consider sex as a covariate in their analyses. However, disaggregated sex and gender information was not collected in our source data. As a result, obtaining consent for sharing individual-level data specific to sex and gender is not applicable to our study. |
|---|---|
| Reporting on race, ethnicity, or other socially relevant groupings | Our study did not use race or ethnic categorisation as conceptual frameworks. Ancestry is reported in the GWAS methods, since this holds significance for the validity of the GWAS. To address population stratification, most of the GWAS included in our study included the principal components as covariate in their analyses. The socially relevant variables in our study were occupational income, household income, and social deprivation. It is important to note that the data used in this study were previously collected within the context of other research, and individual-level data were managed within their respective centers. Household income is the only variable that is dichotomized into different categories, defining UK biobank participants in the following way: (1) low HI: being less than £18,000; (2) low-mid HI: being less than £29,999; (3) mid-high HI: being more than £52,000; (4) high HI: being more than £100,000. This categorisation allowed us to investigate if the effect of income on mental illness is particularly strong at specific income levels. |
| Population characteristics | Not applicable, since no participants were directly enrolled in our study. |
| Recruitment | Not applicable, since no participants were directly enrolled in our study. |
| Ethics oversight | Since our study used summary statistics from previously conducted GWAS, there was no additional ethical approval needed: ethical approval was obtained in all original GWAS studies. |

Note that full information on the approval of the study protocol must also be provided in the manuscript.

# Field-specific reporting

Please select the one below that is the best fit for your research. If you are not sure, read the appropriate sections before making your selection.

☒ Life sciences      ☐ Behavioural & social sciences      ☐ Ecological, evolutionary & environmental sciences

For a reference copy of the document with all sections, see nature.com/documents/nr-reporting-summary-flat.pdf

# Life sciences study design

All studies must disclose on these points even when the disclosure is negative.

| Sample size | To enhance statistical power, the largest available genome-wide association studies were used for both the exposures and the outcomes to perform Mendelian Randomization. The sample sizes ranged from 9,725 to 440,350. Sample size details are reported in Table 2. |
|---|---|
| Data exclusions | No data were excluded from the analyses. |
| Replication | To verify the reproducibility of our findings, we have made the lead SNPs list and summary statistics of the common factor poverty GWAS available, the full GWAS summary statistics available upon reasonable request, and the codes for replicating the analyses accessible. |
| Randomization | Participants were randomly segregated into groups: wild type alleles versus mutant alleles. |

| Blinding | In Mendelian Randomization, estimation of the genetic instrument effect relies on randomization of genotype during meiosis, which is similar to blinding to allocation. |
|---|---|

# Reporting for specific materials, systems and methods

We require information from authors about some types of materials, experimental systems and methods used in many studies. Here, indicate whether each material, system or method listed is relevant to your study. If you are not sure if a list item applies to your research, read the appropriate section before selecting a response.

## Materials & experimental systems

| n/a | Involved in the study |
|---|---|
| ☒ | ☐ Antibodies |
| ☒ | ☐ Eukaryotic cell lines |
| ☒ | ☐ Palaeontology and archaeology |
| ☒ | ☐ Animals and other organisms |
| ☒ | ☐ Clinical data |
| ☒ | ☐ Dual use research of concern |
| ☒ | ☐ Plants |

## Methods

| n/a | Involved in the study |
|---|---|
| ☒ | ☐ ChIP-seq |
| ☒ | ☐ Flow cytometry |
| ☒ | ☐ MRI-based neuroimaging |

