## [Peer Review File · Nature Human Behaviour]

Peer Review Information

Journal: Nature Human Behaviour

Manuscript Title: Investigating the impact of poverty on mental illness in the UK Biobank using Mendelian Randomization

Corresponding author name(s): Mattia Marchi

Reviewer Comments & Decisions:

Decision Letter, initial version:

4th October 2023

Dear Dr Marchi,

Thank you once again for your manuscript, entitled "The impact of poverty on mental illness: Emerging evidence of a causal relationship", and for your patience during the peer review process.

Your Article has now been evaluated by 3 referees. You will see from their comments copied below that, although they find your work of potential interest, they have raised quite substantial concerns. In light of these comments, we cannot accept the manuscript for publication, but would be interested in considering a revised version if you are willing and able to fully address reviewer and editorial concerns.

We hope you will find the referees' comments useful as you decide how to proceed. If you wish to submit a substantially revised manuscript, please bear in mind that we will be reluctant to approach the referees again in the absence of major revisions. We are committed to providing a fair and constructive peer-review process. Do not hesitate to contact us if there are specific requests from the reviewers that you believe are technically impossible or unlikely to yield a meaningful outcome.

In addition to addressing all of the reviewer comments, we ask that you focus on establishing the validity of your MR methods and instruments, as there are strong concerns that the assumptions needed for these analyses to hold cannot be met in your case.

If you wish to submit a suitably revised manuscript, we would hope to receive it within 4 months. I would be grateful if you could contact us as soon as possible if you foresee difficulties with meeting this target resubmission date.

- Include a “Response to the editors and reviewers” document detailing, point-by-point, how you addressed each editor and referee comment. If no action was taken to address a point, you must provide a compelling argument. When formatting this document, please respond to each reviewer comment individually, including the full text of the reviewer comment verbatim followed by your response to the individual point. This response will be used by the editors to evaluate your revision and sent back to the reviewers along with the revised manuscript.

- Highlight all changes made to your manuscript or provide us with a version that tracks changes.

[REDACTED]

Thank you for the opportunity to review your work. Please do not hesitate to contact me if you have any questions or would like to discuss the required revisions further.

Sincerely,

[REDACTED]

REVIEWER COMMENTS:

Reviewer #1:

Remarks to the Author:

The authors use genomic SEM in an attempt to investigate the causal effect of poverty on 9 mental illnesses. They derive a 'multidimensional poverty factor' from household income, occupational income, and social deprivation, and evaluated the putative 'causal' effects using Mendelian randomization. They conclude from these analyses that 'mental illness leads to poverty, and poverty plays a causal role in the development of MDD and SZ, while decreasing the risk of anorexia'. They also find that cognitive abilities explained a significant portion of the impact of poverty on mental illness.

I have a fundamental problem with the concept of the 'genetic architecture of poverty'.

Fully agree with the assertions in the Introduction that (1) there is uncertainty regarding the direction of the association between poverty and mental illness, and (2) that no single indicator can capture the multiple dimensions of poverty. But Mendelian randomization, despite its appeal as a way of circumventing reverse causation and residual confounding, I don't believe is equipped to handle such a complex, multifactorial structural and societal problem as poverty.

First, it is not clear that all three assumptions of MR are met in this context, specifically that (i) the genotype is associated with the exposure; (ii) the genotype is associated with the outcome through the studied exposure only; and (iii) the genotype is independent of other factors which affect the outcome. This needs to be specifically discussed and demonstrated. The only place I can see it mentioned is on P 17 of Methods, 'CA is a likely confounder in the poverty and mental illness relationship, and our instruments for poverty are possibly related to CA (i.e., violation of the independence assumption). ' But how many other possible confounders are there for such a complex societal variable as 'poverty' ? (physical health, dynastic effects, etc.) The conclusions in the abstract are also highly oversimplified, when in fact there is a lot of nuance in the actual results.

For example, the relationship between poverty and ADHD is likely biased from reverse causation and unbalanced pleiotropy. Using separate indicators like household income as a poverty measure, bidirectional effects were found for ADHD and SCZ. The authors state on p 9 that 'This evidence is supporting a vicious circle between poverty and severe mental illness that reiterates'; which of course is highly consistent with many years of epidemiologic observations , but not reflected in the abstract which states that 'poverty plays a causal role in the development of [MDD and] SZ'. Further, there was an (inconsistent across methods) U-shaped relationship between household income and bipolar disorder, which is hard to understand from a genetic causation standpoint.

The authors also conflate educational attainment and cognitive abilities in their discussion, which are of course highly correlated but not the same thing.

Methods- The authors claim in the FAQ that they made a composite measure of poverty that was 'most strongly related to genetic background', but I did not see in the methods how this was determined- merely that these particular 3 indicators were ordered based on factor loadings.

The analyses are indeed very thorough, with over 100 supplementary figures and tables. So the issue is not with comprehensiveness of the analyses, but rather with the starting premise and the interpretation /overall conclusions which vastly oversimplify the findings.

In this case, the restriction of the analysis to strongly European -only ancestry makes sense as it avoids the added complexity of parameters of race/ethnicity; but then again, substantially limits generalizability.

The authors mention in the limitations that the GWAS sample sizes for ANX, ASD, MDD, OCD, and PTSD were relatively small, 'precluding meaningful MR analysis'- and yet, they still don't hold back from making sweeping conclusions about a causal role of poverty in the development of 'mental illness' (although their results only support this direction of effect for MDD). Also, how does one interpret an inverse causal relationship between poverty and anorexia – that anorexia causes one to be wealthier?

I would question the authors' conclusion about the relevance of this study , e.g. "targeted interventions aimed at addressing poverty as a cause of mental illness will advance health equity". We already know (and have known for decades) that poverty plays a significant role in mental illness, for multiple reasons. So this now puts a genetic/biological 'spin' on such patterns , which – for reasons outlined above- does not seem like the right approach to these data.

The Discussion of the manuscript is quite well written and nuanced, addressing highly relevant issues of vertical pleiotropy, dynastic effects, psychosocial mechanisms, and health characteristics that additionally confound these relationships- but this really isn't conveyed in the abstract and the 'FAQ' segment, nor is the manuscript designed to actually address these mechanisms.

Reviewer #2:

Remarks to the Author:

This is a very interesting manuscript which uses Mendelian Randomization to examine the causal effects of poverty on nine mental illnesses. The authors using genomic structural equation modeling to combine GWASs of household income, occupational income, and social deprivation to derive a common "poverty" factor. This is an important step as none of the individual components adequately capture poverty. Using multiple MR methods, the authors find causal interrelationships between poverty and mental illness. Most notably, they find evidence for poverty causing mental illness, a finding that potentially has far reaching implications.

Although this is a potentially controversial topic, the manuscript is very responsibly written, provides appropriate context and sensibly discusses the results. The last bit of the discussion, where the

limitations and weaknesses are discussed, is probably the most important part of this paper. Importantly, the manuscript acknowledges dynastic effects and also acknowledges that these findings may be indexing horizontal pleiotropy. It would be easy to place this paper in the literature without these caveats, but the authors have smartly added these.

A few notes and critiques:

The genetic correlation between latent poverty and cognitive ability was very high -- 0.74
Hard to reconcile this with – It is hard to reconcile this with the last sentence of the results - “However, by utilizing a general factor of poverty the resulting increase in statistical power facilitated the discovery of genetic effects acting to link poverty to mental health independent of the effects of CA.” The authors should clarify this. The authors also state that CA is a confounder (line 585) – it is probably not a confounder but rather part of the causal pathway.

The ADHD findings should be noted in the abstract.

The authors made the decision to limit loci included in genetic instruments to those SNPs that were genome-wide significant. The authors also note that the sample sizes (and consequently power) are small for most of the psychiatric disorders. The authors should reconsider the decision to limit loci (SNPs) to those that reach a very small p-value, since this is a function of power and leads to genetic instruments which account for a small proportion of the variance in some of the traits/disorders.

I also noted some typos:

Line 74 “MR method” should be “The MR method”

Line 77 “Such a study on poverty come with” should be “Such a study on poverty comes with”

Line 90 a closed parenthesis is missing

Line 175 “which provides better” should be “which provides a better”

Line 433 “this study findings”

Reviewer #3:

Remarks to the Author:

The paper conducts an extensive MR analysis to investigate the causal relationship between poverty and nine mental health conditions. Using genetic data from the UK Biobank (UKB), the authors detect effects of poverty indicators (household income, occupational income and social deprivation) on a range of mental illnesses, but also evidence of an inverse relationship. The effects of poverty on mental health seems to be explained to a large extent by individuals’ cognitive ability.

The paper is well written, methodologically sound and concerns a very important topic. I have only a few comments before accepting it for publication.

1) One potential limitation of the authors' work is its susceptibility to selection bias. UK Biobank participants are on average healthier and live in less socioeconomically deprived areas than the UK average (Fry et al. 2017). Note that MR studies are as susceptible to this form of bias as other study designs (Gkatzionis & Burgess, 2019); indeed, there is some evidence that selection bias is not uncommon in MR studies using UK Biobank data (Schoeler et al. 2023).

2) The authors conducted reverse MR analyses only for 4 of their 9 mental health conditions. Why was this the case? The authors mention in the discussion that the sample sizes for some of the nine mental illnesses were not large enough, but the sample sizes reported in Table 2 seem reasonably large to me (maybe except OCD).

3) Was the selection of SNPs associated with poverty conducted in the UK Biobank? If so, this could induce a form of bias known as winner's curse bias, leading to overestimation of SNP-exposure associations. Winner's curse bias is usually not a big concern in studies with a sample size as large as the UK Biobank, but this could perhaps be mentioned as a limitation.

4) Have previous MR studies investigated the relationship between poverty and mental health? I must admit I am not an expert in the field but from a quick Google search I was able to find a few existing papers on the topic (Campbell et al. 2022, Andreu-Bernabeu et al. 2023, Liu et al. 2023) and it may be worth comparing the authors' results against these papers.

References:

- Fry A, Littlejohns TJ, Sudlow C, Doherty N, Adamska L, Sprosen T, Collins R, Allen NE. Comparison of sociodemographic and health-related characteristics of UK Biobank participants with those of the general population. *Am J Epidemiol*, 2017;186(9):1026-1034.
- Gkatzionis A, Burgess S. Contextualizing selection bias in Mendelian randomization: how bad is it likely to be? *Int J Epidemiol*. 2019;48(3):691-701.
- Schoeler T, Speed D, Porcu E, et al. Participation bias in the UK Biobank distorts genetic associations and downstream analyses. *Nat Hum Behav*, 2023;7:1216–1227.
- Campbell D, Green MJ, Davies N, et al. Effects of depression on employment and social outcomes: a Mendelian randomisation study. *J Epidemiol Community Health*, 2022;76:563–571.
- Andreu-Bernabeu A, González-Peñas J, Arango C, Díaz-Caneja CM. Socioeconomic status and severe mental disorders: A bidirectional multivariable Mendelian randomisation study. *MedRxiv* 2023; doi:10.1101/2023.05.25.23290516.

- Liu G, Liu W, Zheng X, Li J. The higher the household income, the lower the possibility of depression and anxiety disorder: evidence from a bidirectional Mendelian randomization study. *Front Psychiatry*, 2023;14, doi:10.3389/fpsy.2023.1264174.

Author Rebuttal to Initial comments

We would like to thank the Editor and the Reviewers for taking the time to review our manuscript, their positive assessments of our work, and their thoughtful comments. We have carefully addressed all concerns raised and provide a point-by-point response below. In the point-by-point response, the text underlined corresponds to the additions we made in the manuscript. We also highlighted in tracked changes the most important changes in the manuscript. We believe the manuscript is considerably improved as a result of Reviewers helpful suggestions.

Editor

In addition to addressing all of the reviewer comments, we ask that you focus on establishing the validity of your MR methods and instruments, as there are strong concerns that the assumptions needed for these analyses to hold cannot be met in your case.

Reply:

We thank the Editor for pointing to this important point that we fully support. We have therefore scrutinized the assumptions again. We have improved the analysis by reducing the odds of the Heywood case in the latent factor analysis which led to minor changes in the output data. We are very confident the data and analysis are sound as explained to the Reviewers. We will adopt a fully transparent policy by publishing the data and analysis code with the manuscript.

Changes in the manuscript:

- *To improve emphasis on the methods we used, we would like to change the title as follows:*

“The impact of poverty on mental illness: investigating the evidence of a causal relationship using Mendelian Randomization”

- *On pages 6-7 of the Supplementary File 1, we added further information on the poverty common factor estimation:*

“Common factor poverty multivariable GWAS

GWAS using Genomic-SEM

The GWAS of the common factor of poverty was conducted in GenomicSEM v0.0.5c, which allows to run a multivariate GWAS for which a common factor defined by genetic indicators is regressed on a SNP. This allows for estimation of a set of summary statistics for the common factor that represent the SNP effects on the common factor. Estimation of a common factor GWAS proceeds in 4 steps: (1) prepare the summary statistics of the observed indicators (in our research these were household income [HI],

social deprivation [SD], and occupational income [OI]); (2) run multivariable linkage-disequilibrium score regression (LDSC) across the observed indicators; (3) combine the summary statistics of HI, SD, and OI; and (4) combine the summary statistics and LDSC output to run the poverty common factor GWAS. In the preparation of the summary statistics the alleles across HI, SD, and OI traits were aligned using the HAPMAP3 reference panel, and summary statistics passed quality control by selecting SNPs with minor allele frequency (MAF) >0.01 and INFO >0.9. Then, LDSC across HI, SD, and OI was run on the SNPs present in the 1000G European reference panel, resulting in 1,165,506 SNPs for HI, 1,165,534 SNPs for SD, and 1,177,612 SNPs for OI which are used to compute the genetic correlation matrix. The number of SNPs shared across HI, OI, and SD, that were retained in the final common factor GWAS was 1,173,569. We specified the common factor model to have fixed variance of the common factor to 1 and letting the lavaan package freely estimate the loading of each indicator. This approach protects against the Heywood case, which is the instance when the standardized factor loading exceeds 1 and, therefore, the indicator has a negative residual variance, which would be inappropriate as not possible in the population. This procedure led to ordering the three indicators based on their factor loadings: household income, occupational income, and social deprivation, with each measure accounting for 100%, 88%, and 74% of the genetic variance, respectively. The final step consisted in running a multivariable GWAS combining the LDSC output with the combined HI, SD, and OI summary statistics to determine our specified common factor model. We calculated the effective sample size of the common factor restricting the summary statistics to MAF limits of 10% and 40% using the equation previously published by Mallard et al (2019 - <https://www.biorxiv.org/content/10.1101/603134v1.abstract>), which resulted in 453,689.”

- Include a “Response to the editors and reviewers” document detailing, point-by-point, how you addressed each editor and referee comment. If no action was taken to address a point, you must provide a compelling argument. When formatting this document, please respond to each reviewer comment individually, including the full text of the reviewer comment verbatim followed by your response to the individual point. This response will be used by the editors to evaluate your revision and sent back to the reviewers along with the revised manuscript.
- Highlight all changes made to your manuscript or provide us with a version that tracks changes.

REVIEWER COMMENTS:

Reviewer #1:

Remarks to the Author:

1. The authors use genomic SEM in an attempt to investigate the causal effect of poverty on 9 mental illnesses. They derive a ‘multidimensional poverty factor’ from household income, occupational income, and social deprivation, and evaluated the putative ‘causal’ effects using Mendelian randomization. They

conclude from these analyses that ‘mental illness leads to poverty, and poverty plays a causal role in the development of MDD and SZ, while decreasing the risk of anorexia’. They also find that cognitive abilities explained a significant portion of the impact of poverty on mental illness.

I have a fundamental problem with the concept of the ‘genetic architecture of poverty’. Fully agree with the assertions in the Introduction that (1) there is uncertainty regarding the direction of the association between poverty and mental illness, and (2) that no single indicator can capture the multiple dimensions of poverty. But Mendelian randomization, despite its appeal as a way of circumventing reverse causation and residual confounding, I don’t believe is equipped to handle such a complex, multifactorial structural and societal problem as poverty.

Reply:

We thank the Reviewer for this frank comment that raises conceptual questions about genetic architecture of poverty and the use of MR. While we agree with the Reviewer that there are important questions with respect to the use of these techniques there is broad consensus that these approaches can be informative when executed in the right way and context¹. We therefore have made great effort to improve the contextual framework of this paper and full transparency of the limitations. This has led to changes throughout the manuscript and abstract.

Changes in the manuscript:

- On page 13, in the Discussion section, the text has been amended to read as follows:

This study primarily builds on genetic evidence and therefore warrants caution with respect to the conceptualization of poverty and mental illness. Moreover, whilst poverty and other measures of socioeconomic status are heritable traits, it is very unlikely that there are direct genetic effects.”

- On page 4 of the Supplementary File 3 (FAQ), we added the following question and reply:

“What is meant by the genetic architecture of poverty, and how did you identify genetic variations related to poverty?”

Our study employed genome-wide association studies (GWAS) to identify genetic variations linked to poverty. A GWAS is a type of study that searches for genetic variants that are linked to some outcome: they have been instrumental in understanding associations of genetic variants with various health conditions, such as schizophrenia or cardiovascular disease, and traits, like height and weight. The genetic variants we just mentioned are called Single-Nucleotide Polymorphisms, or “SNPs”. These are points on the DNA—many millions of them—where people tend to differ in which nucleotide (which one of the four chemical “letters” of DNA: A, C, T, and G) is present. For example, 25% of people might have an “A” at that point on their DNA, whereas the remaining 75% have a “G”. This is the most common, though far from the only, way in which people differ in their DNA. A GWAS study lines up all the SNPs a person has and tests the extent to which each one is linked to the outcome of interest. In the case of height, each SNP might contribute a tiny fraction towards explaining, why some people are taller than others – contributing to bone length, say, or growth hormone production.

Our GWAS estimated a common measure of poverty that captured the genetic overlap between our three indices of poverty: household income, occupational income, and social deprivation. Using genetic data to identify poverty risk may appear unconventional since poverty is not traditionally viewed as a biological condition or trait. However, with increasing GWAS sample sizes, there is growing power to detect SNPs associated with multifactorial phenotypes like diabetes, mental illness, or social conditions such as income and poverty. Clearly, the more complex or multifactorial the phenotype is, the less likely is to observe a direct genetic effect. For example, in type II diabetes mellitus, a multifactorial disease, GWAS could identify genetic variants associated not only with the disease itself but also with related individual characteristics like a preference for consuming sweets or maintaining a sedentary lifestyle. In such cases, the genetic variants associated with upstream or downstream factors exhibit pleiotropic effects. This understanding extends to the genetic architecture of poverty, where GWAS aids in unraveling its complex genetic underpinnings, even if direct effects are improbable.”

First, it is not clear that all three assumptions of MR are met in this context, specifically that (i) the genotype is associated with the exposure; (ii) the genotype is associated with the outcome through the studied exposure only; and (iii) the genotype is independent of other factors which affect the outcome. This needs to be specifically discussed and demonstrated. The only place I can see it mentioned is on P 17 of Methods, ‘CA is a likely confounder in the poverty and mental illness relationship, and our instruments for poverty are possibly related to CA (i.e., violation of the independence assumption). ‘ But how many other possible confounders are there for such a complex societal variable as ‘poverty’ ? (physical health, dynastic effects, etc.)

Reply:

We thank the Reviewer for this comment that allows us to clarify how we dealt with Mendelian randomization (MR) assumptions in our study.

In MR, valid instrumental variables (IV) are defined by three assumptions: they are associated with the risk factor (Relevance); they do not have a common cause with the outcome variables (Independence); and that they influence the outcome only through the risk factor (Exclusion restriction). The first assumption (Relevance) is seen to be met in Two-sample MR by showing that the variants used in the formation of the IV attain genome-wide significance of 5×10^{-8} . In our study each SNP used met that criterion and so satisfied the first assumption. Importantly, weak instrument bias in a two-sample MR design will bias causal effect estimates towards the null and away from the identification of any true causal signal ².

It is never possible to prove the second assumption (Independence) holds ^{1,2}. However, to violate the assumptions of the MR analysis (Independence & Relevance), it is not sufficient for SNPs to have pleiotropic effects on other phenotypes, these effects must be horizontally (not vertically) pleiotropic and affect the outcome via mechanisms other than those captured by the IV. This excludes many potential pleiotropic effects, such as vertically pleiotropic causal chain from a SNP to increased brain volume, to

intelligence, to poverty, to psychiatric disorders. Similarly, vertically pleiotropic effects that are downstream of poverty, but upstream of mental health, also will not cause bias in our estimated causal effect of poverty on psychiatric disorders. For example, a causal chain from SNP to intelligence, to health, to poverty. We investigated the robustness of our results using MR-Egger estimator and MR-PRESSO, this will provide an unbiased estimate of the causal effect of poverty on psychiatric disorders if the pleiotropic effects on other traits are independent of the effects of the SNPs on poverty, however, no method can conclusively prove that the assumptions underlying any of these estimators are true.

Changes in the manuscript:

- To improve readers' understanding of our research process, we moved Figure 5 to the beginning of the Results section on page 4 (since the Methods section is at the end of the manuscript in the Journal). The representation of the schematic overview of the research is now standing in Figure 1, as follows:

"The schematic overview of the study is represented in Figure 1."

- In order to ensure a reader is familiar with how vertical and horizontal pleiotropy are treated in MR we have added the following paragraph to our discussion section on pages 15-16 among study limitations:

"Finally, the genetic variants captured by our measure of poverty are likely to have pleiotropic effects⁶⁷. However, to break the assumptions of MR it is not sufficient for the genetic variants in the instrumental variable (IV) to have pleiotropic effects⁶⁸. The genetic variants used as IVs must have horizontally pleiotropic effects mediated via mechanisms other than those captured by poverty. Should genetic variants have vertically pleiotropic effects, e.g., SNP→neuron→intelligence→education→poverty→health→psychiatric disorder, then our MR derived causal estimates will not be biased. Furthermore, should the SNPs affect other phenotypes, but these phenotypes do not affect outcome, then these effects will not bias our MR estimates. It is possible that the genetic variants identified in our poverty GWAS do have horizontally pleiotropic effects, however, it is unclear what mechanisms would mediate such effects (e.g., personality). In the current study we investigate potentially pleiotropic effects using MVMR to examine the role of CA⁶⁹. Future research should use MVMR to investigate the role of other traits that link poverty to mental health outcomes."

Three relevant references have been added as ref 67, 68, and 69.

- In order to address the Reviewer's concerns within our manuscript regarding the assumptions of MR we have made changes to the text. These changes are on page 13, in the Discussion section:

"Building on data of 18 GWAS this study provided support of a causal relationship between poverty and some mental illnesses."

"This study primarily builds on genetic evidence and therefore warrants caution with respect to the conceptualization of poverty and mental illness. Moreover, whilst poverty and other measures of socioeconomic status are heritable traits, it is very unlikely that there are direct genetic effects."

- We agree with the Reviewer that dynastic effects may bias an MR study. We have now rephrased the paragraph on that source of bias in the study limitation section, on page 15 of the manuscript, to put more emphasis:

“Fourth, it is essential to acknowledge the potential influence of the dynastic effect, wherein characteristics transmitted across generations, such as the association between a parent’s genotype and offspring phenotype, is a known source of bias in MR studies as it violates the second assumption (i.e., independence)⁶². Detecting the exact magnitude of bias resulting from this effect is challenging as current sample sizes preclude such analyses today. Future studies examining the causal link between poverty and mental health outcomes in an MR framework will be better placed to assess any influence of dynastic effects on the causal estimates identified. We also advise triangulation of our findings using complementary research methods in future studies.”

One relevant reference has been added as ref 62.

2. The conclusions in the abstract are also highly oversimplified, when in fact there is a lot of nuance in the actual results.

For example, the relationship between poverty and ADHD is likely biased from reverse causation and unbalanced pleiotropy. Using separate indicators like household income as a poverty measure, bidirectional effects were found for ADHD and SCZ. The authors state on p 9 that ‘This evidence is supporting a vicious circle between poverty and severe mental illness that reiterates’; which of course is highly consistent with many years of epidemiologic observations , but not reflected in the abstract which states that ‘poverty plays a causal role in the development of [MDD and] SZ’.

Reply:

We thank the Reviewer for their suggestion to improve the reporting in our abstract, which we agree is a crucial part of the manuscript. We are constrained by the word limit of 150 words, which is rather low to provide a nuanced overview of our study results. This is why we did not report the results of the secondary analyses using specific poverty indices. In the previous version of our abstract, we reported that “Our investigation showed that mental illness leads to poverty, and poverty plays a causal role in the development of MDD and SZ, while decreasing the risk of AN.” As such, we have stated the bi-directional effects the Reviewer highlighted. However, we seek to improve and have deeply revised the abstract pursuing a more comprehensive reporting, while adhering to the requirements of brevity posed by the Journal.

Changes in the manuscript:

- On page 2, the revised version of the abstract reads as follows:

“The causality of the association between poverty and mental illness is still unclear. Leveraging breakthroughs in genetic studies of mental illness and poverty-related traits, we examined evidence of causal relationships between poverty and nine mental illnesses. Genomic structural equation modeling derived a poverty common factor from household income, occupational income, and social deprivation. Subsequently, Mendelian randomization was used to assess causality. Our investigation showed that schizophrenia and ADHD leads to poverty, and poverty causally contributes to major depression and

schizophrenia, while decreasing the risk of anorexia nervosa. Poverty may also contribute to ADHD albeit with uncertainty due to unbalanced pleiotropy. Cognitive ability played a notable role in these effects, suggesting that individuals' skills related to earning potential might be the actual causal factor for mental health. Further investigations of the bi-directional relationships between cognitive ability, poverty, and mental illness are warranted and may potentially inform public mental health policies."

3. Further, there was an (inconsistent across methods) U-shaped relationship between household income and bipolar disorder, which is hard to understand from a genetic causation standpoint.

Reply:

We thank the Reviewer for this input which allows us to better explain our interpretation of this U-shaped relationship between household income levels and bipolar disorder. We agree that this distribution is hard to understand in terms of genetic causation. In addition, this is not consistently replicated across different MR methods and sensitivity analyses, as highlighted in our manuscript. For instance, the inconsistency in results between methods such as WM-MR and MR-PRESSO, along with the detection of significant distortion due to pleiotropy, raises concerns about the robustness of the observed relationship. We have explicitly discussed these results on pages 10 and 11, emphasizing the need for cautious interpretation. Critically, by looking at Supplementary Table 24 it should be noted that the estimate of the causal effect of high household income on bipolar disorder is based on a limited number of genetic instruments (i.e., 2 SNPs), further reducing to 0 SNP after Steiger filtering (Supplementary Table 26). This underscores the need to exercise caution in drawing definitive conclusions from these specific results. Moreover, the MR of occupational income on mental illness identified a causal effect of bipolar disorder on occupational income, arguably reflecting traits associated with bipolar disorder, such as elevated energy levels and reduced need for sleep, that could impact work productivity. This broader, vertical perspective supports the notion that bipolar disorder may causally influence higher income levels rather than the reverse. In summary, we acknowledge the intricacy of the U-shaped relationship and the challenges in attributing it solely to genetic causation. The comprehensive examination of multiple MR methods, sensitivity analyses, and the exploration of different income dimensions contribute to a nuanced understanding of the observed associations. We hope these additional insights address Reviewer's concerns and the changes in the manuscript contribute to the improvement of our interpretation.

Changes in the manuscript:

- On page 11, we added the interpretation of the MR results on the Steiger filtered set, providing additional support to reverse causation. The new version stands as follows: "Interestingly, the effects observed in the main analysis were not confirmed for BD and AN, yielding evidence of reverse causation rather than direct causality for these associations."

4. The authors also conflate educational attainment and cognitive abilities in their discussion, which are of course highly correlated but not the same thing.

Reply:

We appreciate this Reviewer comment and we have scrutinized the manuscript to avoid confusion. In our discussion section, education is mentioned in two paragraphs. These instances are copied in below:

1. *"In that context, MR can best be viewed as a way to approximately randomly assigned heritable traits that give rise to income differences. Two key candidates of such underlying traits are intelligence and personality. For example, intelligence may lead to both educational advantage and socioeconomic success as well as more healthy behaviors and lead to good mental health ²⁷⁻³¹."*

2. *"CA are closely associated with educational attainment and occupational status, which are often regarded as socio-economic status variables relevant to health ^{27,29,55}. Although the genetic relationship between CA and mental illness differs from the genetic relationship between education and mental illness, education and occupation may still serve as potential targets for interventions. Future research should explore strategies aimed at facilitating individuals' participation in education and employment, which may lead to better mental health ^{8,60}."*

In the first instance we are pointing out that educational attainment is a likely consequence of cognitive ability and may be one of the means by which higher cognitive ability translates to higher earnings. In the second instance we are stating that whilst cognitive ability and education are highly correlated both genetically and phenotypically, they are not the same and indeed differ in their genetic relationship with mental health. However, as education is a more suitable target for therapeutic intervention than cognitive ability the role in facilitating educational development in the etiology of mental illness should be considered. For improvement we have edited the text.

Changes in the manuscript:

- On page 14, we have now altered the text to read as follows:

"CA are closely associated with educational attainment and occupational status, which are often regarded as socio-economic status variables relevant to health ^{27,29,55}. However, the genetic relationship between CA and mental illness differs from the genetic relationship between education and mental illness ^{20,56}. Despite the differences between CA and educational attainment, educational attainment is a likely causal factor in CA differences ⁵⁷⁻⁵⁹ and is a more straightforward target for interventions than CA. Thus, future research should explore strategies aimed at facilitating individuals' participation in education, which may lead to better mental health ^{8,60}."

We added four new relevant references as ref 56, 57, 58, and 59.

5. Methods- The authors claim in the FAQ that they made a composite measure of poverty that was ‘most strongly related to genetic background’, but I did not see in the methods how this was determined- merely that these particular 3 indicators were ordered based on factor loadings.

Reply:

The Reviewer’s observation is valid, and we appreciate the opportunity to further clarify the methodology used in creating and evaluating the common poverty measure. In determining this common factor of poverty, we used the structural equation modeling method, implementing a technique that involved fixing the variance of the common factor to 1 and letting the lavaan package^{3,4} freely estimate the loading of each indicator. This approach protects against the Heywood case, which is the instance when the standardized factor loading exceeds 1 and, therefore, the indicator has a negative residual variance, which would be inappropriate as not possible in the population. This procedure led to ordering the three indicators based on their factor loadings: household income, occupational income, and social deprivation, with each measure accounting for 100%, 88%, and 74% of the genetic variance, respectively. Beyond factor loadings, our statement in the FAQ section (on page 4 of Supplementary file 3) “We made a composite measure of poverty that was most strongly related to genetic background” was also supported by an evaluation of the statistics presented in the Supplementary Table 2. These included the Mean Chi², LDSC intercept, and heritability (h²) statistics, providing a comprehensive assessment of the association between genotype and the poverty phenotype. In particular, Mean Chi² measures the overall strength of association between genetic variants and the phenotype of interest (a high Mean Chi² value indicates that there are many genetic variants that are strongly associated with the trait); the Linkage Disequilibrium Score Regression (LDSC) intercept captures the contribution of factors other than polygenicity (such as population stratification) in inflating the estimates of the GWAS (a high LDSC suggests the presence of confounders); the narrow sense heritability (h²) is a measure of the proportion of phenotypic variation that is attributable to genetic variation (a higher h² value indicates a higher heritability). As can be seen in Supplementary Table 2, the Mean Chi² and the LDSC intercept indicated that the poverty factor outperforms single indicators in capturing phenotypic variations. Finally, the heritability of the common factor was higher than HI and SD, and lower than OI, logically reflecting the average heritability of the three indicators.

Changes in the manuscript:

- On page 5 of the Supplementary File 3 (FAQ), the text has been amended to read as follows:
“We found that our common factor of poverty captured the majority of the genetic variance, with household income, occupational income, and social deprivation accounting for 100%, 88%, and 74%, respectively. In addition, the summary statistics used to evaluate the genotype-phenotype association (i.e., the Mean Chi², the LDSC intercept, and the narrow sense heritability) supported a stronger

relationship for the poverty common factor than the single indicators, with the heritability estimate logically reflecting the average heritability of the three indicators.”

6. The analyses are indeed very thorough, with over 100 supplementary figures and tables. So the issue is not with comprehensiveness of the analyses, but rather with the starting premise and the interpretation /overall conclusions which vastly oversimplify the findings.

In this case, the restriction of the analysis to strongly European -only ancestry makes sense as it avoids the added complexity of parameters of race/ethnicity; but then again, substantially limits generalizability.

Reply:

The Reviewer raises an important point, and we agree that the conceptualization of the analyses and its limitations should be clear to the readership.

Our starting premise was that poverty may cause mental illness. The use of genetic data within a Mendelian randomization framework allows for such causal relationships to be investigated in observational data. The idea is not to oversimplify and state that poverty leads to mental illness, but merely present contemporary evidence of the complex relations between poverty, cognitive ability, and mental illness. As such this paper can have an important contribution to the field of public mental health. In addition, we agree with the Reviewer that the use of data from participants of European ancestry may limit generalizability. We stated that in two parts of the limitation section on page 15 (i.e., 2nd and 5th points of the limitations discussion). However, in order to identify independent SNPs for use as instrumental variables it was necessary to remove SNPs that were correlated with the most significant SNP in a region. These correlations between SNPs, known as linkage disequilibrium (LD) vary by ancestry group, therefore it was not possible to include different ethnicities into the same estimates. We added further clarification of the study premises and limitations, including that about generalizability, in the FAQ section.

Changes in the manuscript:

- On page 4 of the Supplementary File 3 (FAQ), we added the following question and reply:

“What is meant by the genetic architecture of poverty, and how did you identify genetic variations related to poverty?”

Our study employed genome-wide association studies (GWAS) to identify genetic variations linked to poverty. A GWAS is a type of study that searches for genetic variants that are linked to some outcome: they have been instrumental in understanding associations of genetic variants with various health conditions, such as schizophrenia or cardiovascular disease, and traits, like height and weight. The genetic variants we just mentioned are called Single-Nucleotide Polymorphisms, or “SNPs”. These are points on the DNA—many millions of them—where people tend to differ in which nucleotide (which one of the four chemical “letters” of DNA: A, C, T, and G) is present. For example, 25% of people might have

an “A” at that point on their DNA, whereas the remaining 75% have a “G”. This is the most common, though far from the only, way in which people differ in their DNA. A GWAS study lines up all the SNPs a person has and tests the extent to which each one is linked to the outcome of interest. In the case of height, each SNP might contribute a tiny fraction towards explaining, why some people are taller than others – contributing to bone length, say, or growth hormone production.

Our GWAS estimated a common measure of poverty that captured the genetic overlap between our three indices of poverty: household income, occupational income, and social deprivation. Using genetic data to identify poverty risk may appear unconventional since poverty is not traditionally viewed as a biological condition or trait. However, with increasing GWAS sample sizes, there is growing power to detect SNPs associated with multifactorial phenotypes like diabetes, mental illness, or social conditions such as income and poverty. Clearly, the more complex or multifactorial the phenotype is, the less likely is to observe a direct genetic effect. For example, in type II diabetes mellitus, a multifactorial disease, GWAS could identify genetic variants associated not only with the disease itself but also with related individual characteristics like a preference for consuming sweets or maintaining a sedentary lifestyle. In such cases, the genetic variants associated with upstream or downstream factors exhibit pleiotropic effects. This understanding extends to the genetic architecture of poverty, where GWAS aids in unraveling its complex genetic underpinnings, even if direct effects are improbable.”

- On page 6 of the Supplementary File 3 (FAQ), we added the following question and reply:

“To whom your results apply?”

Our findings were made in the context of Great Britain and America (that is, the participants were from England, Scotland, Wales, and USA), in the modern era (that is, the participants filled in their questionnaire on their income at some point within the past approximately ten years). In addition, our analysis was restricted to people who described their ethnic background as White. It will be interesting to test the extent to which the poverty-linked genetic variants and the associations with mental illness we have found apply to people of different backgrounds, and those from different countries, cultures, and time periods, but for now, our data cannot speak to them.”

7. The authors mention in the limitations that the GWAS sample sizes for ANX, ASD, MDD, OCD, and PTSD were relatively small, ‘precluding meaningful MR analysis’- and yet, they still don’t hold back from making sweeping conclusions about a causal role of poverty in the development of ‘mental illness’ (although their results only support this direction of effect for MDD).

Reply:

We are grateful to the Reviewer for picking up on this. It is in fact an overstatement of the limitation that only accounts for the investigation of the backward relationships (the impact of these disorders on poverty). The analyses conducted in MR are dependent on genome wide significant SNPs being identified for the exposure variable. For the outcome variable there is no corresponding p value cutoff and so

despite the small sample sizes in some of the psychiatric variables (ANX, ASD, MDD, OCD, and PTSD) these traits were all suitable to use in MR as outcome variables.

Changes in the manuscript:

- On pages 14-15, in order to clarify our meaning and prevent future misinterpretation, the sentence has now been changed to read as follows:

“First, for ANX, ASD, MDD, OCD, and PTSD, GWAS sample sizes were relatively small, precluding their use as exposure variables for investigating the impact of these disorders on poverty. This is due to the insufficient number of SNPs retained following the selection, clumping, and harmonization steps. For the SNPs selection, we used the threshold of $p < 5e-8$, which is recommended by MR guidelines to ensure the validity of the relevance MR assumption. We could have relaxed this threshold to include additional SNPs, but this could introduce bias. Therefore, we refrained from performing those MR analyses and trust that future, more powerful GWAS on these conditions yield more suitable instruments for MR, bridging this gap.”

- On page 13, we rephrased the first sentence of the Discussion as follows:

“Building on data of 18 GWAS this study provided support of a causal relationship between poverty and some mental illnesses.”

- Moreover, the abstract has been deeply revised and the points that could read as potential overstatements were rephrased to be more careful, specific, and conservative.

8. Also, how does one interpret an inverse causal relationship between poverty and anorexia – that anorexia causes one to be wealthier?

Reply:

The inverse genetic relationship (i.e., the genetic variants associated with increases in anorexia correlate with the genetic variants leading to a reduction in poverty) between anorexia and poverty is well known⁵ and has been shown to extend to measures of education⁶ and is present with intelligence but with a much smaller effect size⁶. What the current study adds is that this relationship has a causal component where less poverty/higher income caused an increase in anorexia risk. Whilst the mechanism is unclear and it is beyond the scope of this study to examine each pathway from poverty to mental illness separately, one potential explanation is that some of the symptoms of anorexia, for example perfectionism, are linked to educational success and less poverty (see Figure 1 below). As discussed also with respect to the relationship between bipolar disorder and income classes, our point here is that some psychopathological dimensions may be facilitative of something else that is linked to higher income and so would be within the MR vertical pleiotropy logic.

Figure 1: one of the possible models explaining the relationship between poverty and mental illness, showing the potential vertical pleiotropic effect of psychopathology.

Changes in the manuscript:

- On page 5 of the Supplementary File 3 (FAQ), we added the following question and reply about the psychosocial mechanisms and pleiotropic influences that could explain our findings:

“How poverty and mental illness could be related?”

Although the relationships we found between poverty and some mental illnesses converged across a series of analyses, it is very unlikely there are direct genetic effects. Rather, the genetic relationship between poverty and mental illness is likely to involve psychosocial mechanisms and vertical pleiotropy. Whilst the investigation of each pathway from poverty to mental illness separately is beyond the scope of this study, it is important to acknowledge that some psychopathological features (such as perfectionism in the context of anorexia nervosa, or attention deficit in the context of ADHD, or, in a broader context, the overall individual’s health status) may be facilitative of something else that is linked to income (such as education attainment). In that context, Mendelian randomization can best be viewed as a way to approximately randomly assign heritable traits that give rise to income differences. In the current study, we investigated the role of cognitive ability using multivariable Mendelian randomization. Future research should use a similar approach to investigate the role of other traits that link poverty to mental health outcomes.”

9. I would question the authors’ conclusion about the relevance of this study, e.g. “targeted interventions aimed at addressing poverty as a cause of mental illness will advance health equity”. We already know (and have known for decades) that poverty plays a significant role in mental illness, for multiple reasons. So this now puts a genetic/biological ‘spin’ on such patterns, which – for reasons outlined above- does not seem like the right approach to these data.

Reply:

This point echoes previous points by the Reviewer that have been well taken and addressed. We do agree that wording is of the essence here and have rephrased the sentence underlined by the Reviewer.

The matter of genetic/biological “spin” may need further clarification.

Despite the use of genetic data, Mendelian randomization does not address whether or not a trait is ‘biological’ or ‘genetic’ in origin. Rather MR is one of many methods available to the epidemiologist that

can add to the weight of evidence regarding the relationship between a potential risk factor and the health of a population. In an MR design, genetic data are used as instrumental variables for modifiable risk factors that are associated with health in a population in order to overcome some of the biases (such as unmeasured confounding) that are present in observational data. Such biases, if left present, would preclude statements regarding the causal nature of the relationship between risk factor and outcome. We would disagree with the reviewer that MR is not the correct approach to take to examine bidirectional causality between poverty and mental illness. Both poverty and mental illness have a genetic component that is captured by common genotyped variants and sample sizes are sufficient that multiple independent SNPs have been identified for use as instrumental variables. Furthermore, whilst some attempts have been made to proxy randomized control trials in which money is randomly allocated to members of the population to examine the causal effects of income on physical and mental health⁷, it is not clear how mental illness could be similarly randomized across a population outside of an MR framework.

Changes in the manuscript:

- On page 14, we altered the sentence about the practical implications of our research mentioned by the Reviewer, to read as follows:

“Although individuals' skills and abilities tied to earning capacity may be the variables with the actual causal effect on mental illness, the findings of this study reiterate the need to further unravel the role of poverty and cognitive ability in mental illness and the way to use this insight to advance mental health.”

10. The Discussion of the manuscript is quite well written and nuanced, addressing highly relevant issues of vertical pleiotropy, dynastic effects, psychosocial mechanisms, and health characteristics that additionally confound these relationships- but this really isn't conveyed in the abstract and the 'FAQ' segment, nor is the manuscript designed to actually address these mechanisms.

Reply:

We appreciate the positive comments of the Reviewer and made a large effort to better explain and provide more nuance throughout the manuscript. At the same time, we must also comply with the Journal guidelines which put a word limit of 150 words for the abstract and 5000 words for the main text, which are fully reasonable and respected.

As the Reviewer correctly states in this comment, the Discussion section already provides a nuanced discussion of our study findings. We therefore included in the FAQ section the acknowledgment of vertical pleiotropy, potential psychosocial mechanisms linking poverty and mental illness, and dynastic effect.

Changes in the manuscript:

- On page 4 of the Supplementary File 3 (FAQ), we now acknowledge vertical pleiotropy:

“However, with increasing GWAS sample sizes, there is growing power to detect SNPs associated with multifactorial phenotypes like diabetes, mental illness, or social conditions such as income and poverty. Clearly, the more complex or multifactorial the phenotype is, the less likely is to observe a direct genetic effect. For example, in type II diabetes mellitus, a multifactorial disease, GWAS could identify genetic variants associated not only with the disease itself but also with related individual characteristics like a preference for consuming sweets or maintaining a sedentary lifestyle. In such cases, the genetic variants associated with upstream or downstream factors exhibit pleiotropic effects. This understanding extends to the genetic architecture of poverty, where GWAS aids in unraveling its complex genetic underpinnings, even if direct effects are improbable.”

- On page 5 of the Supplementary File 3 (FAQ), we added the following question and reply about the psychosocial mechanisms and pleiotropic influences that could additionally confound our findings:

“How poverty and mental illness could be related?”

Although the relationships we found between poverty and some mental illnesses converged across a series of analyses, it is very unlikely there are direct genetic effects. Rather, the genetic relationship between poverty and mental illness is likely to involve psychosocial mechanisms and vertical pleiotropy. Whilst the investigation of each pathway from poverty to mental illness separately is beyond the scope of this study, it is important to acknowledge that some psychopathological features (such as perfectionism in the context of anorexia nervosa, or attention deficit in the context of ADHD, or, in a broader context, the overall individual’s health status) may be facilitative of something else that is linked to income (such as education attainment). In that context, Mendelian randomization can best be viewed as a way to approximately randomly assign heritable traits that give rise to income differences. In the current study, we investigated the role of cognitive ability using multivariable Mendelian randomization. Future research should use a similar approach to investigate the role of other traits that link poverty to mental health outcomes.”

- On page 6 of the Supplementary File 3 (FAQ), we added the following question and reply on the dynastic effect:

“Aren’t the associations you found just due to “dynastic effect”?”

This is a rather technical point, but nonetheless important. Put simply, dynastic effect refers to the transmission of socioeconomic or health-related outcomes across generations within a family. This means that the circumstances, advantages, or disadvantages experienced by one generation may influence the outcomes of the next generations. In the context of income and poverty, if one grows up in a family with a history of important income losses or economic debts, it could be that their income levels in adulthood are not related to their skills and abilities tied to their earning capacity, but rather to the transmission of disadvantaged economic conditions from their parents. Importantly, the dynastic effect can be a potential source of bias in Mendelian randomization (MR) studies. MR relies on the random assortment of genetic variants during meiosis, ensuring that the genetic instrument (SNPs used as proxies for exposure) is not associated with confounding factors. However, if there is a dynastic effect, meaning that the genetic variants not only influence the exposure of interest in the individual but also the exposures of their ancestors or descendants, it could violate the assumptions of MR. Addressing

dynastic effects in MR studies may require additional methodological considerations, such as the use of family-based MR designs, or exploring alternative genetic instruments that are less likely to be influenced by dynastic transmission. It's crucial to note that while MR allows for inferences about the direction of relationships and contributes to evidence of causality, it doesn't provide full proof. These studies serve as an essential first step but should be followed by more detailed population studies."

References used to reply to Reviewer 1

1. Sanderson, E. *et al.* Mendelian randomization. *Nature reviews. Methods primers* **2**, (2022).
2. Davies, N. M., Holmes, M. V. & Davey Smith, G. Reading Mendelian randomisation studies: a guide, glossary, and checklist for clinicians. *BMJ* **362**, (2018).
3. Grotzinger, A. D. *et al.* Genomic SEM Provides Insights into the Multivariate Genetic Architecture of Complex Traits. *Nat Hum Behav* **3**, 513 (2019).
4. Rosseel, Y. Lavaan: An R package for structural equation modeling. *J Stat Softw* **48**, 1–36 (2012).
5. Hill, W. D. *et al.* Genome-wide analysis identifies molecular systems and 149 genetic loci associated with income. *Nature Communications* **10**:1 **10**, 1–16 (2019).
6. Hill, W. D. *et al.* A combined analysis of genetically correlated traits identifies 187 loci and a role for neurogenesis and myelination in intelligence. *Molecular Psychiatry* **24**:2 **24**, 169–181 (2018).
7. Cesarini, D., Lindqvist, E., Östling, R. & Wallace, B. Wealth, Health, and Child Development: Evidence from Administrative Data on Swedish Lottery Players. *Q J Econ* **131**, 687–738 (2016).

Reviewer #2:

Remarks to the Author:

This is a very interesting manuscript which uses Mendelian Randomization to examine the causal effects of poverty on nine mental illnesses. The authors using genomic structural equation modeling to combine GWASs of household income, occupational income, and social deprivation to derive a common “poverty” factor. This is an important step as none of the individual components adequately capture poverty. Using multiple MR methods, the authors find causal interrelationships between poverty and mental illness. Most notably, they find evidence for poverty causing mental illness, a finding that potentially has far reaching implications.

Although this is a potentially controversial topic, the manuscript is very responsibly written, provides appropriate context and sensibly discusses the results. The last bit of the discussion, where the

limitations and weaknesses are discussed, is probably the most important part of this paper. Importantly, the manuscript acknowledges dynastic effects and also acknowledges that these findings may be indexing horizontal pleiotropy. It would be easy to place this paper in the literature without these caveats, but the authors have smartly added these.

Reply:

We thank the Reviewer for their positive and encouraging remarks.

A few notes and critiques:

1. The genetic correlation between latent poverty and cognitive ability was very high -- 0.74
Hard to reconcile this with – It is hard to reconcile this with the last sentence of the results - “However, by utilizing a general factor of poverty the resulting increase in statistical power facilitated the discovery of genetic effects acting to link poverty to mental health independent of the effects of CA.” The authors should clarify this. The authors also state that CA is a confounder (line 585) – it is probably not a confounder but rather part of the causal pathway.

Reply:

We thank the Reviewer for their insightful comment which allows us to highlight an important strength of our research, which is the use of a common factor of poverty rather capturing multidimensional aspects of poverty than a single indicator. The high correlation between the common factor poverty and cognitive ability (i.e., 0.74) suggests a substantial shared genetic influence, which can exist alongside independent genetic pathways linking poverty to mental health. Our results indicate that the introduction of a common factor for poverty allows the identification of these independent genetic effects, which is likely due to the gain in statistical power obtained using the latent poverty factor. In addition, we agree that cognitive ability (CA) is part of the causal pathway linking poverty to mental illness. However, given the results of MR of the common factor poverty and CA, and in particular those of the Steiger test, CA is likely an upstream causal factor in the relationship. We have made this point clearer in the last sentence of the Results section and accepted the suggestion to better clarify the role of cognitive ability as part of the causal pathway rather than a confounder in the Methods section.

Changes in the manuscript:

- On pages 12-13, we added clarification of the MVMR findings of common factor poverty and CA vs. mental illness:

“However, by utilizing a general factor of poverty the resulting increase in statistical power facilitated the discovery of genetic effects acting to link poverty to mental health independent of the effects of CA. Therefore, while cognitive ability is correlated with poverty, does not fully account for the observed genetic effects on mental health. These findings, when considered alongside the results of the MR of P

and CA, point to CA fitting as an upstream component in the complex causal pathway linking poverty to mental illness.”

- On page 19, we amended the sentence to clarify that CA is likely part of the causal pathway linking poverty and mental illness. The new sentence reads as follows:

“Finally, we hypothesized that CA is likely involved in the poverty and mental illness causal pathway, and our instruments for poverty are possibly related to CA.”

2. The ADHD findings should be noted in the abstract.

Reply:

We thank the Reviewer for the suggestion to add this part. We agree the ADHD findings are important and have added it to the abstract while pointing out the limitation for this particular analysis.

Changes in the manuscript:

- On page 2, the revised version of the abstract reads as follows:

“The causality of the association between poverty and mental illness is still unclear. Leveraging breakthroughs in genetic studies of mental illness and poverty-related traits, we examined evidence of causal relationships between poverty and nine mental illnesses. Genomic structural equation modeling derived a poverty common factor from household income, occupational income, and social deprivation. Subsequently, Mendelian randomization was used to assess causality. Our investigation showed that schizophrenia and ADHD leads to poverty, and poverty causally contributes to major depression and schizophrenia, while decreasing the risk of anorexia nervosa. Poverty may also contribute to ADHD albeit with uncertainty due to unbalanced pleiotropy. Cognitive ability played a notable role in these effects, suggesting that individuals’ skills related to earning potential might be the actual causal factor for mental health. Further investigations of the bi-directional relationships between cognitive ability, poverty, and mental illness are warranted and may potentially inform public mental health policies.”

3. The authors made the decision to limit loci included in genetic instruments to those SNPs that were genome-wide significant. The authors also note that the sample sizes (and consequently power) are small for most of the psychiatric disorders. The authors should reconsider the decision to limit loci (SNPs) to those that reach a very small p-value, since this is a function of power and leads to genetic instruments which account for a small proportion of the variance in some of the traits/disorders.

Reply:

We thank the Reviewer for their comment, which is in line with a comment of Reviewer 3 (comment #2) and allows us to provide a more comprehensive clarification on this matter. For ANX, ASD, MDD, OCD, and PTSD, the number of SNPs selected for the reverse MR analyses (i.e., to investigate the impact of

these disorders on poverty) was insufficient. The number of genome-wide significant SNPs is a function of the GWAS power, which is partly a function of its sample size. In this situation, we had two options: (i) relax the selection criteria to include enough SNPs to perform the MR analyses, e.g., lowering the threshold for SNP selection by increasing the p-value from 5-08 to higher values; or (ii) refrain from performing the reverse MR analyses for those phenotypes. The first option would imply a violation of the first MR assumption (relevance), potentially introducing bias. Still being the most conservative, the second option limits the comprehensiveness of our results. We carefully discussed the pros and cons of the two options within the research group. Considering the research topic and the potential risk of bias linked to the inclusion of less relevant SNPs (e.g., the risk of including SNPs with strong pleiotropic effects), we opted for the second option, which in our opinion is striking the best balance between methodological rigorousness and coherence. We revised the limitation section of the manuscript, to improve the description of this limit and clarify our choice.

Changes in the manuscript:

- On pages 14-15, among the study limitations, we added the following:

“First, for ANX, ASD, MDD, OCD, and PTSD, GWAS sample sizes were relatively small, precluding their use as exposure variables for investigating the impact of these disorders on poverty. This is due to the insufficient number of SNPs retained following the selection, clumping, and harmonization steps. For the SNPs selection, we used the threshold of $p < 5e-8$, which is recommended by MR guidelines to ensure the validity of the relevance MR assumption. We could have relaxed this threshold to include additional SNPs, but this could introduce bias. Therefore, we refrained from performing those MR analyses and trust that future, more powerful GWAS on these conditions yield more suitable instruments for MR, bridging this gap.”

4. I also noted some typos:

Line 74 “MR method” should be “The MR method”

Line 77 “Such a study on poverty come with” should be “Such a study on poverty comes with”

Line 90 a closed parenthesis is missing

Line 175 “which provides better” should be “which provides a better”

Line 433 “this study findings”

Reply:

We thank the Reviewer for their careful check. We fixed all the typos they pointed out.

Changes in the manuscript:

- On page 3, “MR method” now reads as:

“The MR method”

- On page 3, “Such a study on poverty come with”, now reads as:

“Studies on poverty come with the challenge of defining poverty.”

- On page 3, the parenthesis has been closed. Now it reads as:

“(HI)”

- On page 7, “which provides better”, now reads as:

“which provides a better”

- On page 14, “this study findings”, now reads as:

“the findings of this study”

Reviewer #3:

Remarks to the Author:

The paper conducts an extensive MR analysis to investigate the causal relationship between poverty and nine mental health conditions. Using genetic data from the UK Biobank (UKB), the authors detect effects of poverty indicators (household income, occupational income and social deprivation) on a range of mental illnesses, but also evidence of an inverse relationship. The effects of poverty on mental health seems to be explained to a large extent by individuals’ cognitive ability.

The paper is well written, methodologically sound and concerns a very important topic. I have only a few comments before accepting it for publication.

Reply:

We thank the Reviewer for their positive and encouraging remarks.

1) One potential limitation of the authors’ work is its susceptibility to selection bias. UK Biobank participants are on average healthier and live in less socioeconomically deprived areas than the UK average (Fry et al. 2017). Note that MR studies are as susceptible to this form of bias as other study designs (Gkatzionis & Burgess, 2019); indeed, there is some evidence that selection bias is not uncommon in MR studies using UK Biobank data (Schoeler et al. 2023).

Reply:

We thank the Reviewer for this thoughtful comment. We agree this is an important consideration that has not been acknowledged in the previous version of the manuscript. However, the references the Reviewer cited point out that selection bias is an important concern when the selection effect is large, or when the aim of the study is to derive generalizable prevalence and incidence rates of disease. We have reasons to state that the selection effect in our study is low since our study does not (i) assess the causal effect of a risk factor on secondary disease or disease progression (e.g., considering the effect of body mass index on breast cancer progression, to be included in an analysis of disease progression, a

participant must have had an initial disease event. If BMI is a risk factor for breast cancer risk, then selection into the sample population would be a function of a collider, namely BMI, and hence bias would occur); (ii) assess the causal effect of a risk factor on a disease outcome in an elderly population, which is susceptible of survival bias; (iii) assess a causal effect in a subpopulation (e.g., the females). In addition, there is evidence that UK Biobank is sufficiently large and heterogeneous to provide valid scientific inferences of associations between exposures and health conditions that are generalizable to other populations, which is in line with the aim of our research.

We added the discussion of this potential selection bias in the limitation section, highlighting also that its impact is likely to be less than other biases (e.g., pleiotropic or dynastic effects) in our study.

Changes in the manuscript:

- On page 15, we added the following to the study limitations:

“Fifth, it should be noted that previous research pointed out that UK Biobank participants are on average healthier and live in less socioeconomically deprived areas than the UK average⁶³. This could introduce potential selection bias in MR studies using UK Biobank data, inflating the risk of Type I error^{64,65}. However, the impact of selection bias is particularly strong when the selection effect is large, for instance, when studying disease progression, secondary diseases, or specific subpopulations, such as elderly people⁶⁴. In the context of our research, the selection effect is likely to be low. Furthermore, there is evidence that UK Biobank is sufficiently large and heterogeneous to provide valid scientific inferences of associations between exposures and health conditions, which is in line with the aim of our research⁶³. Therefore, the impact of selection bias in our study is likely to be less than other sources of bias, such as pleiotropic or dynastic effects.”

- The three relevant references the Reviewer suggested have been added as ref 63, 64, and 65.

2) The authors conducted reverse MR analyses only for 4 of their 9 mental health conditions. Why was this the case? The authors mention in the discussion that the sample sizes for some of the nine mental illnesses were not large enough, but the sample sizes reported in Table 2 seem reasonably large to me (maybe except OCD).

Reply:

We thank the Reviewer for their comment, which is in line with a comment of the Reviewer 2 and allows us to provide a more comprehensive clarification on this matter. For ANX, ASD, MDD, OCD, and PTSD, the number of SNPs selected for the reverse MR analyses was insufficient. The number of genome-wide significant SNPs is a function of the GWAS power, which is partly a function of its sample size. In this situation, we had two options: (i) relax the selection criteria to include enough SNPs to perform the MR analyses, e.g., lowering the threshold for SNP selection by increasing the p-value from 5-08 to higher values; or (ii) refrain from performing the reverse MR analyses for those phenotypes. The first option would imply a violation of the first MR assumption (relevance), potentially introducing bias. Still being

the most conservative, the second option limits the comprehensiveness of our results. We carefully discussed the pros and cons of the two options within the research group. Considering the research topic and the potential risk of bias linked to the inclusion of less relevant SNPs (e.g., the risk of including SNPs with strong pleiotropic effects), we opted for the second option, which in our opinion is striking the best balance between methodological rigorousness and coherence. We revised the limitation section of the manuscript, to improve the description of this limit and clarify our choice.

Changes in the manuscript:

- On pages 14-15, in the limitation section, we added the following:

“First, for ANX, ASD, MDD, OCD, and PTSD, GWAS sample sizes were relatively small, precluding their use as exposure variables for investigating the impact of these disorders on poverty. This is due to the insufficient number of SNPs retained following the selection, clumping, and harmonization steps. For the SNPs selection, we used the threshold of $p < 5e-8$, which is recommended by MR guidelines to ensure the validity of the relevance MR assumption. We could have relaxed this threshold to include additional SNPs, but this could introduce bias. Therefore, we refrained from performing those MR analyses and trust that future, more powerful GWAS on these conditions yield more suitable instruments for MR, bridging this gap.”

3) Was the selection of SNPs associated with poverty conducted in the UK Biobank? If so, this could induce a form of bias known as winner’s curse bias, leading to overestimation of SNP-exposure associations. Winner’s curse bias is usually not a big concern in studies with a sample size as large as the UK Biobank, but this could perhaps be mentioned as a limitation.

Reply:

We thank the Reviewer for their comment. We agree that winner’s curse may be bias in SNPs selection of the poverty factor, although as the Reviewer correctly acknowledges a sample size as large as the UK Biobank should have mitigated this potential distortion. We accepted the suggestion to mention this potential limitation in the limitation section of our manuscript.

Changes in the manuscript:

- On page 15, we added the following to the study limitations:

“Sixth, the selection of SNPs associated with the latent factor poverty was performed in the UK Biobank, leading to the potential risk of overestimation of the SNP-phenotype association due to winner’s curse bias. However, the impact of this bias is generally not substantial in large samples, as the UK Biobank⁶⁶.”

- We added one relevant reference, as ref 66.

4) Have previous MR studies investigated the relationship between poverty and mental health? I must admit I am not an expert in the field but from a quick Google search I was able to find a few existing papers on the topic (Campbell et al. 2022, Andreu-Bernabeu et al. 2023, Liu et al. 2023) and it may be worth comparing the authors' results against these papers.

Reply:

We thank the Reviewer for suggesting these valuable references. However, one of these (Andreu-Bernabeu et al. 2023) is still a pre-print and we would refrain from citing papers that have still not completed the peer-review process. Nevertheless, we eagerly accepted the suggestion to add a comparison of our findings with those from Campbell et al. 2022, Liu et al. 2023 to the discussion, and we also found another MR study of potentially modifiable risk factors for depression, including household income.

Changes in the manuscript:

- On page 13, we added mention to previous MR studies of poverty and mental illness. The new paragraph now reads as:

“These findings complement previous evidence of social inequalities in mental health across different countries, sample sizes, and study designs ^{2,7,11,21,22}, including previous MR studies focusing on MDD, household income, and unemployment ²³⁻²⁵. However, our research adds evidence that the relationship between poverty and mental illness is valid across different measures of poverty and extends the investigation to a wider range of mental illnesses.

- We added three relevant references, as ref 23, 24, and 25.

References:

- Fry A, Littlejohns TJ, Sudlow C, Doherty N, Adamska L, Sprosen T, Collins R, Allen NE. Comparison of sociodemographic and health-related characteristics of UK Biobank participants with those of the general population. *Am J Epidemiol*, 2017;186(9):1026-1034.
- Gkatzionis A, Burgess S. Contextualizing selection bias in Mendelian randomization: how bad is it likely to be? *Int J Epidemiol*. 2019;48(3):691-701.
- Schoeler T, Speed D, Porcu E, et al. Participation bias in the UK Biobank distorts genetic associations and downstream analyses. *Nat Hum Behav*, 2023;7:1216–1227.
- Campbell D, Green MJ, Davies N, et al. Effects of depression on employment and social outcomes: a Mendelian randomisation study. *J Epidemiol Community Health*, 2022;76:563–571.
- Andreu-Bernabeu A, González-Peñas J, Arango C, Díaz-Caneja CM. Socioeconomic status and severe mental disorders: A bidirectional multivariable Mendelian randomisation study. *MedRxiv* 2023; doi:10.1101/2023.05.25.23290516.

- Liu G, Liu W, Zheng X, Li J. The higher the household income, the lower the possibility of depression and anxiety disorder: evidence from a bidirectional Mendelian randomization study. *Front Psychiatry*, 2023;14, doi:10.3389/fpsy.2023.1264174.

Decision Letter, first revision:

1st May 2024

Dear Dr. Marchi,

Thank you for your patience as we've prepared the guidelines for final submission of your Nature Human Behaviour manuscript, "The impact of poverty on mental illness: investigating the evidence of a causal relationship using Mendelian Randomization" (NATHUMBEHAV-23072192A). Please see the below note:

Dear Dr Marchi,

Please accept my sincere apologies once again for the extraordinary delay in sharing our final requests on your manuscript. I have now gone through your materials and very much appreciate your efforts during the revision process. In finalizing your manuscript, please do take into consideration the outstanding reviewer comments we shared with you.

Additionally, I have edited your manuscript file and your very informative FAQ, which I attach to this message. The main aim of my comments/suggestions on your manuscript file is to improve clarity and ensure that descriptions of your findings stay as close as possible to the evidence presented (this is especially important given the sensitivity of the topic and the potential for misuse of your findings). My suggestions on your excellent FAQ are primarily aimed to make the text more concise as it is intended for non-specialist readers.

Finally, I have provided some additional requests in the attached checklist (which also includes requests by our Editorial Assistant).

Please don't hesitate to contact me if you have any questions. Given the substantial delay, I will make sure to prioritize your manuscript when you submit the finalized version.

Please carefully follow the step-by-step instructions provided in the attached file, and add a response in each row of the table to indicate the changes that you have made. Please also check and comment on any additional marked-up edits we have proposed within the text. Ensuring that each point is addressed will help to ensure that your revised manuscript can be swiftly handed over to our production team.

Nature Human Behaviour offers a Transparent Peer Review option for new original research manuscripts submitted after December 1st, 2019. As part of this initiative, we encourage our authors to support increased transparency into the peer review process by agreeing to have the reviewer comments, author rebuttal letters, and editorial decision letters published as a Supplementary item. When you submit your final files please clearly state in your cover letter whether or not you would like to participate in this initiative. Please note that failure to state your preference will result in delays in accepting your manuscript for publication.

In recognition of the time and expertise our reviewers provide to Nature Human Behaviour's editorial process, we would like to formally acknowledge their contribution to the external peer review of your manuscript entitled "The impact of poverty on mental illness: investigating the evidence of a causal relationship using Mendelian Randomization". For those reviewers who give their assent, we will be publishing their names alongside the published article.

Cover suggestions

We welcome submissions of artwork for consideration for our cover. For more information, please see our guide for cover artwork.

ORCID

Non-corresponding authors do not have to link their ORCIDs but are encouraged to do so. Please note that it will not be possible to add/modify ORCIDs at proof. Thus, please let your co-authors know that if they wish to have their ORCID added to the paper they must follow the procedure described in the following link prior to acceptance: <https://www.springernature.com/gp/researchers/orcid/orcid-for-nature-research>

Nature Human Behaviour has now transitioned to a unified Rights Collection system which will allow our Author Services team to quickly and easily collect the rights and permissions required to publish your work. Approximately 10 days after your paper is formally accepted, you will receive an email in providing you with a link to complete the grant of rights. If your paper is eligible for Open Access, our Author Services team will also be in touch regarding any additional information that may be required to arrange payment for your article.

Please note that *Nature Human Behaviour* is a Transformative Journal (TJ). Authors may publish their research with us through the traditional subscription access route or make their paper immediately open access through payment of an article-processing charge (APC). Authors will not be required to make a final decision about access to their article until it has been accepted. Find out more about Transformative Journals

Please use the following link for uploading these materials:
[REDACTED]

Best regards,
[REDACTED]

On behalf of

[REDACTED]

Reviewer #1:

Remarks to the Author:

I appreciate the thorough and detailed responses provided by the authors.

In addition, I appreciate their transparent approach, as they include the data and analysis code with the manuscript. Given the controversial (and easily misunderstood) content of this manuscript, it is particularly important that the analysis is rigorous and transparent as possible. The authors have made efforts to mitigate the potential for misunderstandings, but I still think further contextualization of the findings is important.

The manuscript is much improved but I still have a fundamental problem with conducting genomic structural equation modeling on the variables of household income, occupational income, and social deprivation (the three indices of poverty). All of these, but social deprivation in particular- as a measure of poverty in the area in which one lives- seems inseparable from dynastic effects and thus violates the assumptions of MR. The authors do address this concern to some extent in their excellent Discussion.

A few specific suggestions/ concerns remain.

1. Title- The revised title is definitely an improvement, but please consider further revising to : ‘The impact of poverty on mental illness in the UK Biobank: investigating the evidence of a causal relationship using Mendelian Randomization’ or at the very least, The impact of poverty on mental illness in the UK: “ . Generalizability of findings on poverty based on this relatively healthy White British sample to other populations is a big question.
2. The FAQ for a lay audience is helpful, but frankly I still think there is a lot of potential for misunderstanding. To make this clearer, I suggest that on p 4 of Supplementary File 3 “What is meant by the genetic architecture of poverty, and how did you identify genetic variations related to poverty?” I suggest you clarify “ Our study employed genome-wide association studies (GWAS) to identify genetic variations linked to poverty in a white British middle-aged adult sample”.
3. I suggest in the discussion the authors cite this highly relevant paper from Abdellaoui et al 2019 , ‘Genetic correlates of social stratification in Great Britain’, also in the UKBB which shows significant geographic clustering at the genetic level, likely reflecting SES-driven migration effects. Notably, there are also substantial changes in polygenic scores by birth year, highlighting the complexity of these ‘genetic’ associations.

4. The discussion first paragraph implies the findings more general than they really are, e.g. "These findings complement previous evidence of social inequalities in mental health across different countries, sample sizes, and study designs .." It is worth reiterating here that these findings are in a white British sample.

Reviewer #3:

Remarks to the Author:

I am happy with the authors' changes to my previous comments and recommend that their manuscript is accepted for publication.

Author Rebuttal, first revision:

We would like to thank the Editor and the Reviewers for taking the time to review our manuscript, their positive assessments of our work, and their thoughtful comments. We have carefully addressed all concerns raised and provide a point-by-point response below. In the point-by-point response, the text underlined corresponds to the additions we made in the manuscript. We also highlighted in tracked changes the most important changes in the manuscript. We believe the manuscript is considerably improved as a result of Reviewers helpful suggestions.

Editor

Please accept my sincere apologies once again for the extraordinary delay in sharing our final requests on your manuscript. I have now gone through your materials and very much appreciate your efforts during the revision process. In finalizing your manuscript, please do take into consideration the outstanding reviewer comments we shared with you.

Additionally, I have edited your manuscript file and your very informative FAQ, which I attach to this message. The main aim of my comments/suggestions on your manuscript file is to improve clarity and ensure that descriptions of your findings stay as close as possible to the evidence presented (this is especially important given the sensitivity of the topic and the potential for misuse of your findings). My suggestions on your excellent FAQ are primarily aimed to make the text more concise as it is intended for non-specialist readers.

Finally, I have provided some additional requests in the attached checklist (which also includes requests by our Editorial Assistant).

Please don't hesitate to contact me if you have any questions. Given the substantial delay, I will make sure to prioritize your manuscript when you submit the finalized version.

Reply:

We understand the challenges of the editorial team and are grateful that the manuscript was moved forward. It will tremendously help the younger team members in their careers. We thank the Editor for their clear and comprehensive comments. Below, we have provided a point-by-point reply to the Reviewer's comments. Additionally, we have revised the manuscript and the FAQ according to the Editor's suggestions. We have also completed the checklists to address all requests. Replies to the editors' comments insofar relevant are also in the tracked change manuscript. We sincerely appreciate your cooperation during the revision process and are looking forward to the forthcoming steps.

REVIEWER COMMENTS:

Reviewer #1:

I appreciate the thorough and detailed responses provided by the authors.

In addition, I appreciate their transparent approach, as they include the data and analysis code with the manuscript. Given the controversial (and easily misunderstood) content of this manuscript, it is particularly important that the analysis is rigorous and transparent as possible. The authors have made efforts to mitigate the potential for misunderstandings, but I still think further contextualization of the findings is important.

The manuscript is much improved but I still have a fundamental problem with conducting genomic structural equation modeling on the variables of household income, occupational income, and social deprivation (the three indices of poverty). All of these, but social deprivation in particular- as a measure of poverty in the area in which one lives- seems inseparable from dynastic effects and thus violates the assumptions of MR. The authors do address this concern to some extent in their excellent Discussion.

Reply:

We do understand the Reviewer's concern here. However, whilst we do acknowledge that dynastic effects may have influenced our results (line 471 on page 15) it is important to note that our primary analysis was conducted on our general factor of poverty. This poverty factor was derived using the variance that is common across each indicator of poverty meaning if dynastic effects were the sole or main signal captured by social deprivation they would not be included in this general factor.

A few specific suggestions/ concerns remain.

1. Title- The revised title is definitely an improvement, but please consider further revising to : "The impact of poverty on mental illness in the UK Biobank: investigating the evidence of a causal relationship using Mendelian Randomization" or at the very least, The impact of poverty on mental illness in the UK: ". Generalizability of findings on poverty based on this relatively healthy White British sample to other populations is a big question.

Reply:

We understand the Reviewer's concern, which is further emphasized by the Editor's proposed title, as indicated in the marked-up edit in the manuscript file.

Changes in the manuscript:

- On the title page, the new title stands as follows:

"Investigating the impact of poverty on mental illness in the UK Biobank using Mendelian Randomization".

2. The FAQ for a lay audience is helpful, but frankly I still think there is a lot of potential for misunderstanding. To make this clearer, I suggest that on p 4 of Supplementary File 3 “What is meant by the genetic architecture of poverty, and how did you identify genetic variations related to poverty?” I suggest you clarify “ Our study employed genome-wide association studies (GWAS) to identify genetic variations linked to poverty in a white British middle-aged adult sample”.

Reply:

We do understand the Reviewer’s point here and thank them for the suggestion which has been accepted.

Changes in the manuscript:

- On the Supplementary File, page 152, FAQ section, we added the clarification suggested by the Reviewer. The amended text reads as follows:

“Our study employed genome-wide association studies (GWAS) to identify genetic variations linked to poverty in a white British adult sample.”

3. I suggest in the discussion the authors cite this highly relevant paper from Abdellaoui et al 2019 , ‘Genetic correlates of social stratification in Great Britain’, also in the UKBB which shows significant geographic clustering at the genetic level, likely reflecting SES-driven migration effects. Notably, there are also substantial changes in polygenic scores by birth year, highlighting the complexity of these ‘genetic’ associations.

Reply:

The Reviewer raises an interesting point. Regarding the effect of birth year, what Abdelloui et al find is that a polygenic score for education is greater in older participants (Fig 5a in Abdelloui et al, 2019). Furthermore, once participants were split into those born in coal mining regions and those born outside of coal mining regions, polygenic scores for education were consistently greater in those born outside of coal mining regions (Fig 5b in Abdelloui et al, 2019). There was also a significant interaction between birth year and coal region where the slope is greater for those within coal regions indicating a steeper decline of polygenic score for education across time. Additionally, when participants were placed in four groups: moved away from coal field, moved to coal field, stayed in coal field, stayed out of coal field, those that moved from coal fields had the highest polygenic load for education (Fig 5c in Abdelloui et al, 2019).

The decline in polygenic score for education across time (Fig 5a in Abdelloui et al, 2019) and between coal regions and non-coal regions (Fig 5b in Abdelloui et al, 2019) is consistent with previous work showing that education is associated with longevity. The significant interaction between birth year and

coal region (Fig 5b in Abdelloui et al, 2019), and the finding that those who were born in but moved from coal fields have the highest polygenic load for education is indicative of a genetic effect. In this instance, Abdelloui et al show that genetic inheritance can drive migration around the UK where those who have the heritable traits linked to educational success move away from low SES environments. As such, the results of our study complement those of Abdelloui et al as we find that poverty is one of the causal factors in some psychiatric disorders. Part of this causal chain may very well be that those with the heritable traits linked to lower levels of poverty move away from stressful environments/jobs/neighborhoods and so place themselves at a lower risk of disorder. We have now amended the manuscript to include reference to the Abdelloui et al paper and including these genetic migration events as one possible mechanism by which poverty is linked with psychiatric disorder.

Changes in the manuscript:

- On page 14 (Discussion section) we added the following text:

"The causal relationship between poverty and mental illness is likely to involve material, psychological, behavioral, and biological pathways. For example, those with the heritable traits linked to lower levels of poverty may be more able to move away from stressful environments/jobs/neighborhoods and so place themselves at a lower risk of disorder³⁵. Furthermore, the level of development of the welfare system may be a material mediator to the health-damaging effects of income losses^{36,37}."

- The Abdelloui et al, 2019 study has been cited as reference #35.

4. The discussion first paragraph implies the findings more general than they really are, e.g. "These findings complement previous evidence of social inequalities in mental health across different countries, sample sizes, and study designs .." It is worth reiterating here that these findings are in a white British sample.

Reply:

We thank the Reviewer for their comment, which highlights the need to enhance the clarity of our discussion and minimize potential overstatements. We agree that the poverty data used to estimate the causal effect on mental illness were obtained from a predominantly white British sample (i.e., the UK Biobank [UKBB]), but it should also be clear that our study assessed the inverse effect of mental illness on poverty based on the mental illness data sourced from the Psychiatric Genomics Consortium (PGC), excluding UKBB to avoid sample overlap in the TwoSample MR analyses. As a result, the data is not exclusively white-British but includes individuals of broad European and American ancestry. Therefore, we describe the target population of our findings as individuals of Western ancestry, as we now state in the revised first paragraph of the discussion.

Changes in the text:

- On page 13 the first paragraph of the discussion has been amended to read as follows:

“Our findings, in two Western ancestry samples, complement the available evidence on social inequalities in mental health collected across different countries, samples, and study designs^{2,7,11,21,22}, including previous MR studies focusing on MDD, household income, and unemployment²³⁻²⁵.”

Final Decision Letter:

Dear Dr Marchi,

We are pleased to inform you that your Article "Investigating the impact of poverty on mental illness in the UK Biobank using Mendelian Randomization", has now been accepted for publication in *Nature Human Behaviour*.

Please note that *Nature Human Behaviour* is a Transformative Journal (TJ). Authors may publish their research with us through the traditional subscription access route or make their paper immediately open access through payment of an article-processing charge (APC). Authors will not be required to make a final decision about access to their article until it has been accepted. Find out more about Transformative Journals

Authors may need to take specific actions to achieve compliance with funder and institutional open access mandates. If your research is supported by a funder that requires immediate open access (e.g. according to Plan S principles) then you should select the gold OA route, and we will direct you to the compliant route where possible. For authors selecting the subscription publication route, the journal's standard licensing terms will need to be accepted, including self-archiving policies. Those licensing terms will supersede any other terms that the author or any third party may assert apply to any version of the manuscript.

With best regards,

[REDACTED]